# ON/OFF domains shape receptive field structure in mouse visual cortex

Elaine Tring[1], Konnie K. Duan[2] & Dario L. Ringach[1,3 ✉]

In higher mammals, thalamic afferents to primary visual cortex (area V1) segregate according to their responses to increases (ON) or decreases (OFF) in luminance. This organization induces columnar, ON/OFF domains postulated to provide a scaffold for the emergence of orientation tuning. To further test this idea, we asked whether ON/OFF domains exist in mouse V1. Here we show that mouse V1 is indeed parceled into ON/OFF domains. Interestingly, fluctuations in the relative density of ON/OFF neurons on the cortical surface mirror fluctuations in the relative density of ON/OFF receptive field centers on the visual field. Moreover, the local diversity of cortical receptive fields is explained by a model in which neurons linearly combine a small number of ON and OFF signals available in their cortical neighborhoods. These findings suggest that ON/OFF domains originate in fluctuations of the balance between ON/OFF responses across the visual field which, in turn, shapes the structure of cortical receptive fields.

[1] Department of Neurobiology, David Geffen School of Medicine, UCLA, Los Angeles, CA 90095, USA. [2] Harvard-Westlake School, Studio City, CA 91604, USA. [3] Department of Psychology, UCLA, Los Angeles, CA 90095, USA. ✉email: dario@ucla.edu

The clustering of geniculate afferents by ON/OFF types in the primary visual cortex is a ubiquitous feature of the thalamocortical projection[1–4]. Such organization induces columnar, ON/OFF cortical domains, where the responses of neurons are dominated either by the onset (ON) or offset (OFF) of luminance within their receptive fields[5]. Initially described in mink[3] and ferret[4], ON/OFF domains have now been documented in cat[2] and monkey[6]. The importance of ON/OFF domains is that their spatial organization appears linked to the structure of receptive fields and cortical maps[2,5,7–16]. Horizontal electrode penetrations through the primary visual cortex reveal cortical sites alternatively dominated by ON and OFF responses, with simple cells, having adjacent ON/OFF subregions, located between the peaks of ON and OFF domains[5]. Moreover, the cortical maps for orientation preference, direction, and retinal disparity, have a strong relationship to the spatial organization of ON/OFF domains[5]. Are ON/OFF domains necessary for orientation tuning to emerge? What is their origin? Here we tackle these questions by analyzing the distribution of ON and OFF kernels in populations of cortical neurons of mouse primary visual cortex[17–21].

If ON/OFF domains are required for the generation of orientation tuning, one would expect mice to possess them as well. Indeed, we find that mouse V1 is parceled into ON/OFF domains. Moreover, the average of simple-cell receptive fields in a population correlates with the difference between the average of mono-contrast ON and OFF receptive fields, a result that replicates a related finding in cat[8]. We introduce a simple model of ON/OFF domain formation that postulates a biased input from the geniculate, where ON or OFF inputs dominate in different parts of the visual field. We confirm the model's prediction that fluctuations in the relative density ON/OFF neurons on the cortical surface, which define ON/OFF domains, mirror fluctuations (biases) in the relative density of ON/OFF receptive field centers on the visual field. Finally, we show how the local diversity in the two-dimensional structure of simple-cell receptive fields, is explained by a model in which neurons linearly combine a handful of ON and OFF signals available within their cortical neighborhoods[16,22]. Altogether, these findings support the view that ON/OFF domains reflect biases of the local representation of ON/OFF signals from the geniculate which, in turn, constrains and shapes the structure of cortical receptive fields.

## Results

We used two-photon imaging in alert, head-fixed mice to measure the responses of neurons in a volume of the primary visual cortex to visual stimuli. At the beginning of each session, we conducted a coarse retinotopic mapping by splitting the microscope's field of view into a 3 by 3 grid, averaging the raw fluorescence signals within each region, and computing the responses evoked by stimulation with a small, flickering checkerboard from different locations across the visual field (Fig. 1a, Methods). This allowed us to verify that the sign of the retinotopy was correct for V1[23] and to measure the center of the aggregate receptive field on the visual field, which averaged 24 deg in azimuth on the right visual hemifield and 5 deg in elevation across the experiments. Then, we used sparse-noise stimulation (Fig. 1b, Methods) to record the responses of neurons within a cortical volume sampled with 4–9 optical sections spaced 30 μm apart (Fig. 1c). A standard data analysis pipeline comprised of image registration, cell segmentation, signal extraction, and deconvolution steps, yielded the estimated spiking of neurons (Fig. 1d). The centroid of the regions of interest (ROIs), along with the depth of the optical section, allowed us to assign each neuron a coordinate in cortical space, $(x_1, x_2, x_3)$ (Fig. 1c).

We computed the ON and OFF receptive fields (or kernels) of each segmented cell in the population by correlating their responses with the locations of bright and dark patches in the stimuli at different time delays (Fig. 2a, left panels, Methods). We defined ON/OFF kernels at the optimal delay for which the norm of the kernel attained its maximum value. Cells that had only a statistically significant ON kernel were defined as ON cells (Fig. 2b, top panel); a similar definition was applied to OFF cells. We refer to such cells as "mono-contrast" because they only respond to either an increment or a decrement in luminance. Cells with both ON and OFF kernels could be split into simple or complex depending on the degree of spatial overlap between the kernels (Fig. 2b, c)[24,25]. Across all the experiments, OFF cells were the most numerous representing 55% of the population, while ON cells accounted for 27% of the population, and the remaining 18% comprised cells with both ON and OFF responses (Methods, table). This latter group was composed of 50.5% of simple cells and 49.5% of complex cells. To estimate the center locations $(y_1, y_2)$ of the kernels in visual space we fit two-dimensional Gaussians (Fig. 2b, right panels). In simple cells, where the peaks of the ON and OFF kernels are displaced in space, the difference between the ON and the OFF kernels showed flanking subregions of opposite signs (Fig. 2b, bottom panel).

**Mouse V1 is parceled into ON/OFF domains**. To test for the existence of ON/OFF domains we computed the difference in the density of ON and OFF mono-contrast neurons in native cortical space. Given a set of points $(x_1^i, x_2^i)$ where ON cells are located on the cortical surface (ignoring depth) we estimated their density via a kernel estimate, $f_{on}(x_1, x_2) = 1/N_{on} \sum_i G(x_1 - x_1^i, x_2 - x_2^i)$, where $N_{on}$ is the number of ON cells and $G(\cdot)$ is a Gaussian kernel[26] (Methods). A similar density estimate can be obtained for OFF cells, as $f_{off}(x_1, x_2) = 1/N_{off} \sum_i G(x_1 - x_1^i, x_2 - x_2^i)$. The fluctuation in the density of ON and OFF cells is given by the difference $f_{on}(x_1, x_2) - f_{off}(x_1, x_2)$. We observe that $f_{on}(x_1, x_2)$ and $f_{off}(x_1, x_2)$ typically have non-uniform distributions that peak at different locations (Fig. 2d). In regions where the density of ON cells peaks, we see a trough in the density of OFF cells (e.g., Fig. 2d, red circle). Similarly, in regions where the density of OFF cells peak, we observe a trough in the density of ON cells (for example, Fig. 2d, blue circles). As a result of this relationship, the difference $f_{on}(x_1, x_2) - f_{off}(x_1, x_2)$ shows a clear spatial structure with regions where the density of ON cells is higher than OFF cells (ON domains), and regions where the density of OFF cells is higher than ON cells (OFF domains). The statistical significance of these fluctuations was assessed by Monte Carlo simulations where the ON and OFF labels of the cells are randomly shuffled, allowing us to determine the locations where the deviations attained a significance at the 0.001 level (Fig. 2d, right panel, blue and red level sets). These results are typical of our datasets, all of which showed regions with statistically significant ON and OFF dominance (Supplementary Fig. 1). When ON/OFF maps were calculated based on the top half or bottom half optical sections, we found they correlated significantly in all but one case (Methods, table). This is consistent with the idea ON/OFF domains form columnar structures. The areas of the ON and OFF domains, defined as the region enclosed by the levels sets, yield medians sizes for ON domains of 5750 [2652, 9670] μm² and slightly larger OFF domains with 8520 [4190, 16700] μm² (median, [25%,75%] percentiles). We conclude that mouse V1 contains ON/OFF domains.

**A biased-input model of ON/OFF domains**. What could be the origin of ON/OFF domains? One possibility is that the relative

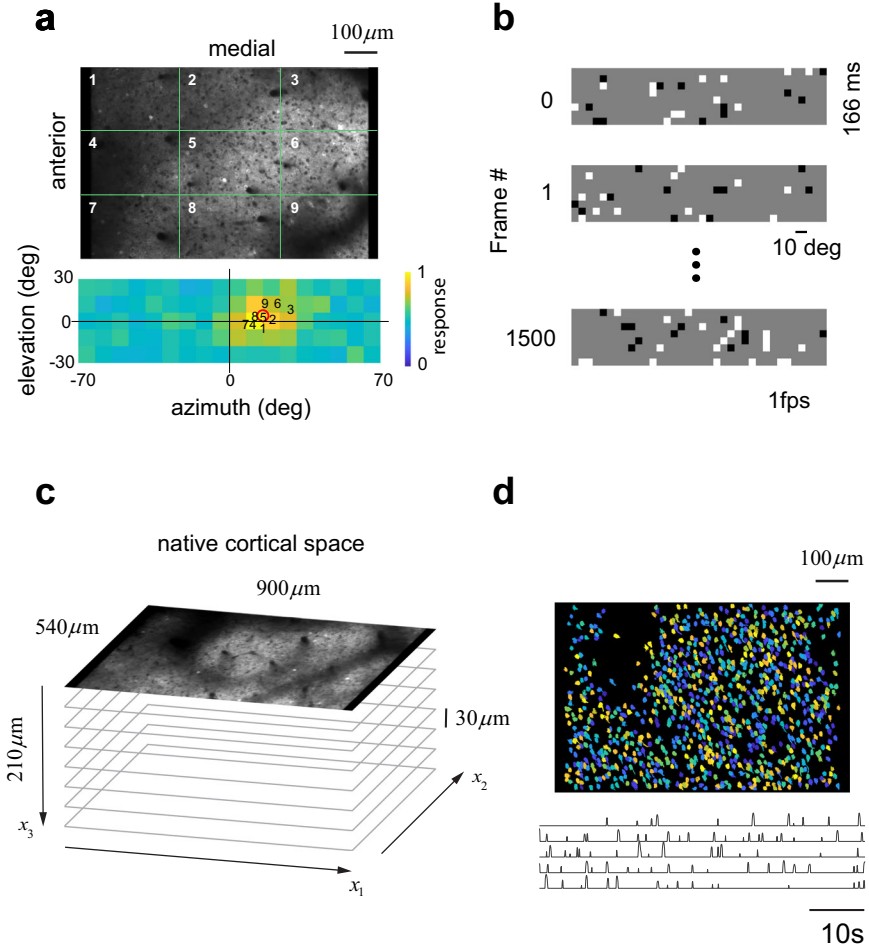

**Fig. 1 Basic methods. a** Coarse retinotopic mapping. The average signal originating from nine sectors defined by a 3 × 3 grid on the field of view of the microscope were used to map the aggregate receptive field and determine its center. The image at the bottom shows the location of the centers for each sector superimposed on top of the aggregate receptive field for the entire population (normalized to its peak). **b** Sparse-noise stimulus. Images were flashed for 166 ms and presented at a rate of 1 per second on a wide field screen. **c** Volumetric sampling in primary visual cortex (see Methods for details on the volumes for each experiment). **d** Segmentation of regions of interest (ROIs) and five sample traces showing spike inference from calcium signals. For visualization, ROIs are assigned random colors.

density of ON/OFF receptive field centers on the visual field conveyed by the geniculate input varies across the visual field. If the retinotopic mapping between visual and cortical space is sufficiently accurate, such fluctuations would be mirrored in the cortex, generating corresponding ON/OFF domains (Fig. 3a). According to such a biased-input model, ON/OFF domains simply mirror a property of the distribution of ON and OFF receptive fields centers on the visual field. Therefore, such an arrangement predicts a correlation between the location of ON/OFF domains and the relative balance of ON/OFF receptive field centers represented by the geniculate population.

If ON and OFF signals are unevenly represented across the visual field, their spatial distribution could influence the structure of cortical receptive fields[2,11,27–30]. For example, the wiring of simple cells requires the selection of ON and OFF inputs within each of their corresponding subregions[24,31,32]. The outcome of Hebbian competition between ON and OFF inputs[33] could be decided in favor of one or the other depending on the initial balance between them. If ON signals are dominant in one location of the visual field, it is likely that simple cells will develop an ON subregion at that location (and similarly for OFF signals). Thus, the uneven representation of ON/OFF signals at the input could induce corresponding cortical domains that determine the

location in the visual space of ON and OFF subregions in simple cells.

It is also reasonable to speculate that cortical neurons can only sample from thalamic signals that fall within a restricted volume centered around their positions in the cortex (Fig. 3b, circles). Cells with access to inputs dominated by either ON or OFF geniculate inputs would tend to develop mono-contrast receptive fields (Fig. 3b, windows *a* and *b*), either slightly elongated (if pooling more than one input) or circular (if pooling just one input). In both cases, their receptive fields closely reflect those of the inputs. We will assume we can take the receptive fields of mono-contrast cells as a proxy for those conveyed by the geniculate. In contrast, cortical cells located in neighborhoods with access to both ON and OFF signals can develop receptive fields with adjacent subregions of the opposite polarity, expressing a diversity of profiles depending on the selected inputs[27,30] (Fig. 3b, window *c*). Consider the case shown by window *c* in Fig. 3b. Pooling from the two inputs labeled 1 would generate a simple cell with an orientation preference of 45 deg, while pooling from the inputs labeled 2 would result in a RF with an orientation preference of 135 deg. Pooling from all available inputs would generate a vertically oriented RF (Fig. 3b, *all*). Importantly, given the set of ON/OFF receptive fields depicted, it is impossible to construct a horizontally tuned receptive field, as there are no pairs

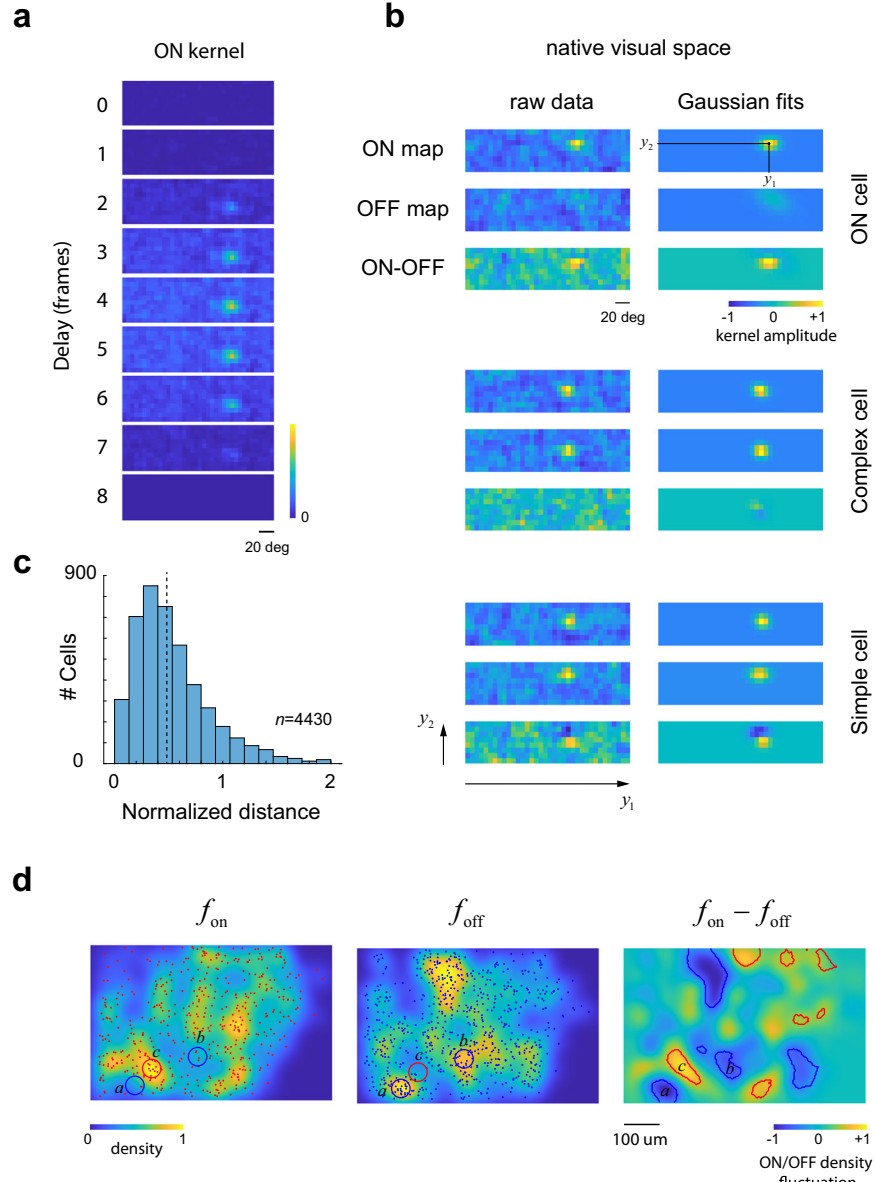

**Fig. 2 Calculation of ON/OFF kernels and domains. a** Example of a spatio-temporal ON kernel. The panels show the correlation between the response of a neuron and the location of the presentation of bright stimuli across the visual field for different delay times between stimulus and response. The peak response, as measured by the norm of the kernel, occurs 4 frames (260 ms) after stimulus onset. Kernel is normalized between zero and one. **b** Raw receptive field measurements (left panels) and their Gaussian fits (right panels). Top, Cell with only a significant ON kernel. Middle, a complex cell with largely overlapping ON and OFF kernel. Bottom, a simple cell with spatially displaced ON and OFF kernels. ON and OFF kernels are normalized jointly to the absolute value attained by either of them, thus the colormap ranges from 0 to 1 and the colormap in panel **a** applies. In the case of ON-OFF difference maps, we normalized by the maximum absolute value of the map, thus the colormap ranges from −1 to +1. **c** Distribution of normalized distance in V1 neurons with significant ON and OFF maps in $n = 4430$ neurons pooled across all experiments. Normalized distance is defined as the distance between receptive field centers divided by the average sigma of the Gaussian fit. We define simple cells as those with a normalized distance larger than one half. **d** Demonstration of ON/OFF domains in native cortical space. The image on the left shows the distribution of ON cells on the cortical surface (we are projecting depth away) along with a pseudo-colormap showing the estimated density. The density estimation for OFF cells appears in the middle panel. Note that both densities appear to peak at different locations. Blue circles show two peak locations for the density of OFF cells. The red circle shows a peak location for ON cells. ON/OFF domains are evident in the difference of the densities, as shown in the right image. Level sets depict areas where the fluctuations exceed what might be expected by chance at a 0.001 level by randomly shuffling the ON/OFF labels of the cells (without changing their positions). Source data provided for panel **c**.

of ON/OFF inputs with receptive fields centers displaced vertically. Thus, a biased input constrains the shape of the receptive fields the cortex can develop[27]. Moreover, under mild conditions, the model predicts the average receptive field of simple cells in a cortical volume should correlate with the average of the ON/OFF signals within its neighborhood[27,30,34].

**Alignment of visual and cortical spaces via canonical correlation analysis.** To test the predictions of the model we first bring cortical and visual fields into alignment using canonical correlation analysis[35], which yields two linear transformations of the data. Cortical space is mapped to $\hat{x}_1 = a_{11}(x_1 - \bar{x}_1) + a_{12}(x_2 - \bar{x}_2) + a_{13}(x_3 - \bar{x}_3)$ and $\hat{x}_2 = a_{21}(x_1 - \bar{x}_1) + a_{22}(x_2 - \bar{x}_2) + a_{23}(x_3 - \bar{x}_3)$,

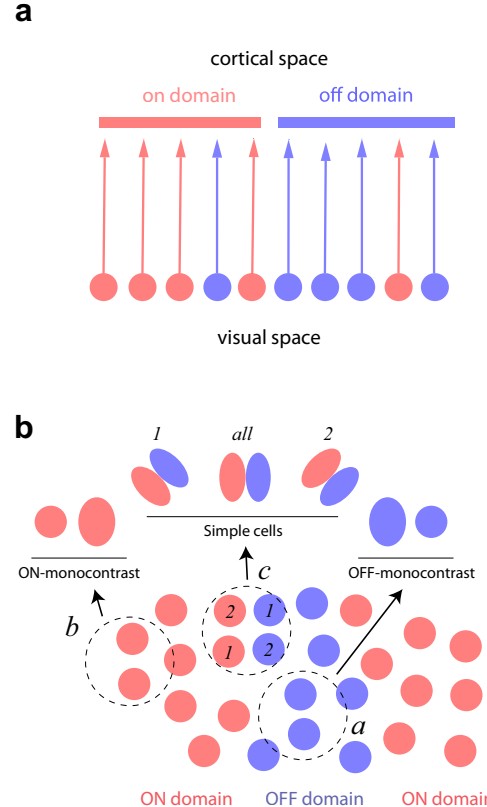

**Fig. 3 A biased-input model of ON/OFF domains. a** The cartoon depicts a 1D version of the biased-input model. The bottom layer shows the location and polarities of ON/OFF cells in the visual field (ON=red, OFF=blue). The top layer shows the input of those signals into the cortex assuming an accurate retinotopy. Here, we readily see that fluctuations in the relative density of ON and OFF cells ought to follow a corresponding fluctuation in the density of ON and OFF receptive field locations in the visual field. Thus, ON/OFF domains could arise from a property of the input. Thalamic projections, of course, include axonal arborizations that are not depicted in this diagram to avoid clutter. **b** The cartoon represents a top view from the cortex of a set of geniculate ON and OFF-center inputs dominating the representation in different parts of the visual field. We assume an accurate retinotopy, so the spatial distribution of ON/OFF receptive fields also represents the distribution of ON/OFF afferents into the cortex. We hypothesize that clusters of ON inputs establish cortical ON domains, and clusters of OFF inputs define OFF domains. Dashed circles represent neighborhoods over which three different cortical cells can sample the geniculate inputs. Such areas are determined by the extent of thalamic arborizations and the size of cortical dendritic trees. Cortical cells with access just to (*a*) OFF or (*b*) ON inputs will develop mono-contrast receptive fields with a single subregion of the corresponding polarity. Cortical cells with access to both types of signals (*c*) can develop receptive fields with two subregions (three possible examples are illustrated).

or in matrix form $\hat{x} = (x - \bar{x})A$. Similarly, the visual field is mapped by $\hat{y}_1 = b_{11}(y_1 - \bar{y}_1) + b_{12}(y_2 - \bar{y}_2)$ and $\hat{y}_2 = b_{21}(y_1 - \bar{y}_1) + b_{22}(y_2 - \bar{y}_2)$, or in matrix form $\hat{y} = (y - \bar{y})B$. The transformations maximize the correlations between the pairs $(\hat{x}_1, \hat{y}_1)$ and $(\hat{x}_2, \hat{y}_2)$ while ensuring the orthogonality of $(\hat{x}_1, \hat{y}_2)$ and $(\hat{x}_2, \hat{y}_1)$, and equalizing the variance of all canonical variables to one. The inclusion of cortical depth $(x_3)$ in the dataset allowed us to compensate for slight departures of the objective from the surface normal, as the maximum correlation between canonical cortical and visual space variables occurs when the matrix $A$ projects the location of the neurons to a plane parallel to the cortical surface. The linear

transformations preserve the major features of the distributions of ON and OFF cells in the cortex. Locations where OFF cells are dominant in native cortical space, for example, will remain an OFF dominant location in canonical space (Fig. 4a, d, asterisks). Similarly, a location in the visual field where we see a dominance of OFF receptive fields centers will remain so in canonical visual space (Fig. 4b, d, asterisks).

The outcome of canonical correlation analysis is a representation of each mono-contrast neuron in the population by its canonical coordinates in cortical space $(\hat{x}_1, \hat{x}_2)$ (Fig. 4c) and its canonical coordinates in visual space $(\hat{y}_1, \hat{y}_2)$ (Fig. 4d). In this example, the correlation between the first pair of canonical variables, $\hat{x}_1$ and $\hat{y}_1$, was $\rho = 0.93$, while the correlation between the second pair, $\hat{x}_2$ and $\hat{y}_2$, was $\rho = 0.81$ (Fig. 4e, f). Incidentally, if we perform canonical correlation analysis separately for ON and OFF cells, the correlation between the first and second canonical variables is higher for OFF than ON cells ($p < 0.001$, paired sign-rank test, two-tailed), consistent with earlier reports that ON cells tend to have larger retinotopic scatter than OFF cells[5,28]. With this representation of the data at hand, we are now in the position to study the fluctuation of the density of ON/OFF cells in the canonical cortical domain, the fluctuation in the density of ON/OFF receptive fields centers in the canonical visual field, and how these two functions relate to each other.

**Testing predictions of the biased-input model.** We used kernel density techniques to estimate the probability distribution of ON and OFF cells in canonical cortical space, denoted by $f_{on}^{\hat{x}}$ and $f_{off}^{\hat{x}}$, respectively, (Methods). Similarly, we estimated the probability distribution of ON and OFF receptive fields centers in canonical visual space, yielding $f_{on}^{\hat{y}}$ and $f_{off}^{\hat{y}}$. To detect fluctuations in the spatial distribution of ON and OFF cells in canonical cortical space, we calculated the difference $f_{on}^{\hat{x}} - f_{off}^{\hat{x}}$. Similarly, to detect fluctuations in the distribution of ON and OFF receptive field centers in canonical visual space, we calculated the difference $f_{on}^{\hat{y}} - f_{off}^{\hat{y}}$ (Fig. 5).

Reflecting our prior observations in native cortical space, we see that the distributions of $f_{on}^{\hat{x}}$ and $f_{off}^{\hat{x}}$ tend to be patchy, peaking in different locations, which results in the difference $f_{on}^{\hat{x}} - f_{off}^{\hat{x}}$ having statistically significant peaks and troughs (Fig. 5, top rows of each panel). We assessed the likelihood that the observed magnitudes in the fluctuations of $f_{on}^{\hat{x}} - f_{off}^{\hat{x}}$ could arise by chance in Monte Carlo simulations where ON/OFF labels were randomly shuffled (Methods). Level sets were computed corresponding to the $p = 0.001$ significance level (Fig. 5, red and blue solid curves). Thus, as expected, we observe ON/OFF domains in the transformed canonical space as well.

Similarly, we can calculate the density of receptive field centers for ON ($f_{on}^{\hat{x}}$) and OFF ($f_{off}^{\hat{x}}$) mono-contrast cells in our populations, as well as their fluctuations, $f_{on}^{\hat{y}} - f_{off}^{\hat{y}}$ (Fig. 5, bottom rows of each panel). These data corroborate a prediction of the biased-input model, fluctuations in $f_{on}^{\hat{x}} - f_{off}^{\hat{x}}$ are robustly mirrored by fluctuations in the balance of ON/OFF receptive field centers on the canonical visual field, $f_{on}^{\hat{y}} - f_{off}^{\hat{y}}$. This is shown by the significant correlation between these functions (Fig. 5, inset). Statistical significance was established by computing the likelihood that the observed level of correlation could arise by chance in controls that randomly shuffled ON/OFF labels (Fig. 5, $p$-values, Methods).

**ON/OFF domains shape receptive field structure.** A second prediction of the model is that the average receptive field of simple cells should correlate with the difference between the

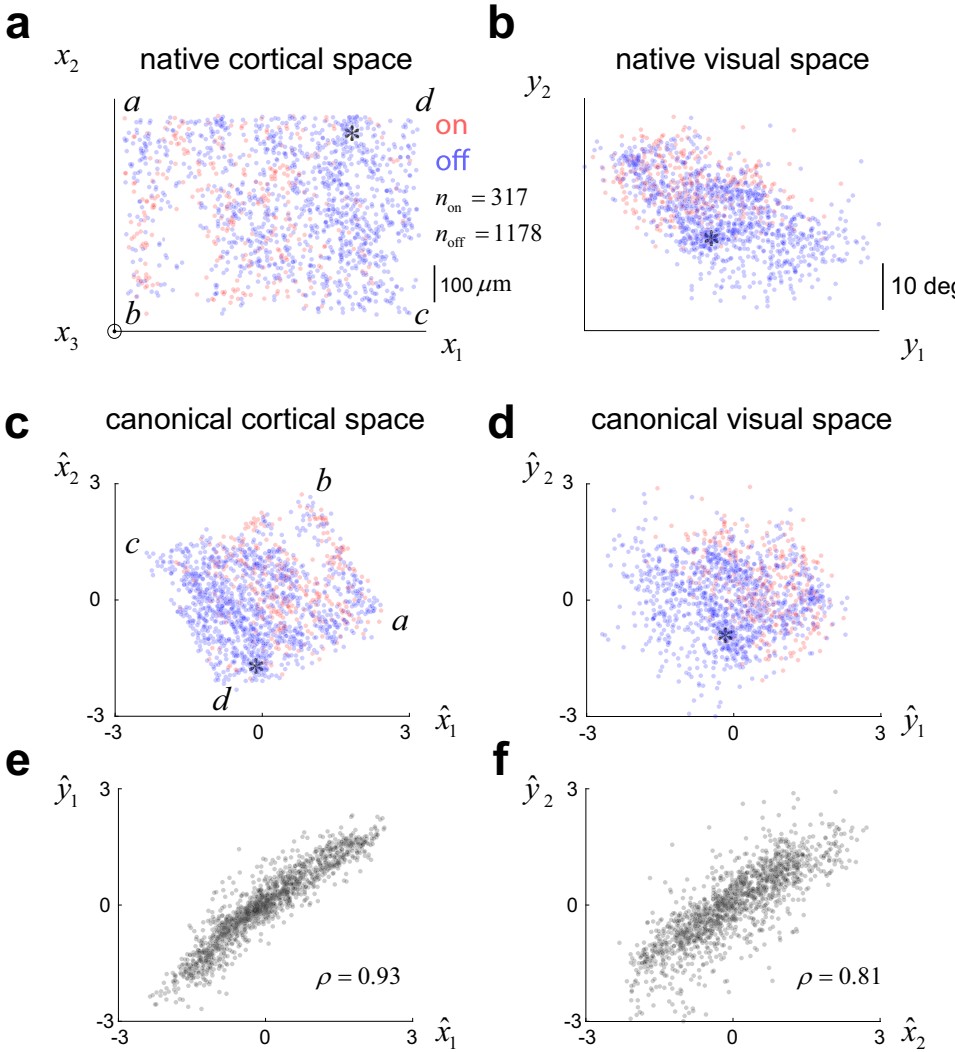

**Fig. 4 Alignment of cortical and visual space via canonical correlation analysis. a**, **b** Distribution of ON (red) and OFF (blue) neurons in the native cortical and visual spaces ($n_{on} = 317$, $n_{off} = 1178$). Native cortical space has an $x_3$ component that comes out of the page, so here the data are projected into the $(x_1, x_2)$ plane ignoring depth. **c**, **d** Same data as in **a**, **b** now represented in canonical cortical and visual spaces. Note how the locations $a$, $b$, $c$, $d$, corresponding to the corners of the imaging plane, are mapped to canonical visual space. A location dominated by OFF cells in cortical or visual space (**a**, **b** asterisks) will map to a location dominated by OFF cells in canonical cortical or canonical visual space (**c**, **d** asterisks). **e** Correlation between the first pair of canonical variables $\hat{x}_1$ and $\hat{y}_1$ and **f** the second pair $\hat{x}_2$ and $\hat{y}_2$. Source data provided for all panels.

average ON and OFF signals. Let us denote the average receptive field of ON cells (in native visual space) across the population by $\mu_{on}$ and adopt a corresponding definition for $\mu_{off}$. For simple cells, we define their receptive field as the difference between their ON and OFF kernels. Finally, the average, linear receptive field of the simple-cell population is denoted by $\mu_s$. The data show a strong, significant correlation between $\mu_{on} - \mu_{off}$ and $\mu_s$ ($p < 10^{-10}$ in all cases) (Fig. 6a). As predicted, the aggregate, linear receptive field of simple cells matches the difference between the average of ON and OFF receptive fields.

It is worth emphasizing that despite the agreement between the averages $\mu_s$ and $\mu_{on} - \mu_{off}$, there is substantial variability in the structure of simple-cell receptive fields in the population (Fig. 6b). This can be observed by computing the distribution of the correlation coefficient between the receptive fields of individual neurons and the average $\mu_s$ (Fig. 6c). The variability in the population is reflected in the spread of the correlation coefficient which spans a range from $-0.72$ to $0.88$. The mean of the distribution, of course, is significantly larger than zero (sign-rank test, $p < 10^{-10}$), as anticipated from the correlation between the

averages (Fig. 6a). This pattern of results was typical of all our datasets.

Can the variability in the local distribution of ON and OFF cells account for the diversity of simple-cell receptive field structure? To test this idea, we modeled the receptive field of simple cells as a non-negative, linear combination of the receptive fields of its $k$-nearest ON and OFF cells (Fig. 6d). To select the size of the neighborhood (the parameter $k$), we calculated how the goodness-of-fit of the model as a function of $k$ and found that it begins to saturate at $k \sim 5$ (Fig. 6e). The performance of the model for a choice of $k = 5$ was very good: the correlation coefficient between the fits and the actual receptive fields was highly skewed towards one with an average of $\bar{\rho} = 0.84$ (Fig. 6f, g). Cells in the local neighborhood were no more than $\sim 50\,\mu m$ away from the target cell (Fig. 6h). Interestingly, there was a large disparity in the weights estimated for the cells in the neighborhood (Fig. 6i). When the weights of ON and OFF cells are plotted in descending order, it is evident that their distribution is sparse. On average, more than 90% of the total synaptic input could be accounted for the largest two inputs for both ON and OFF neurons (Fig. 4i,

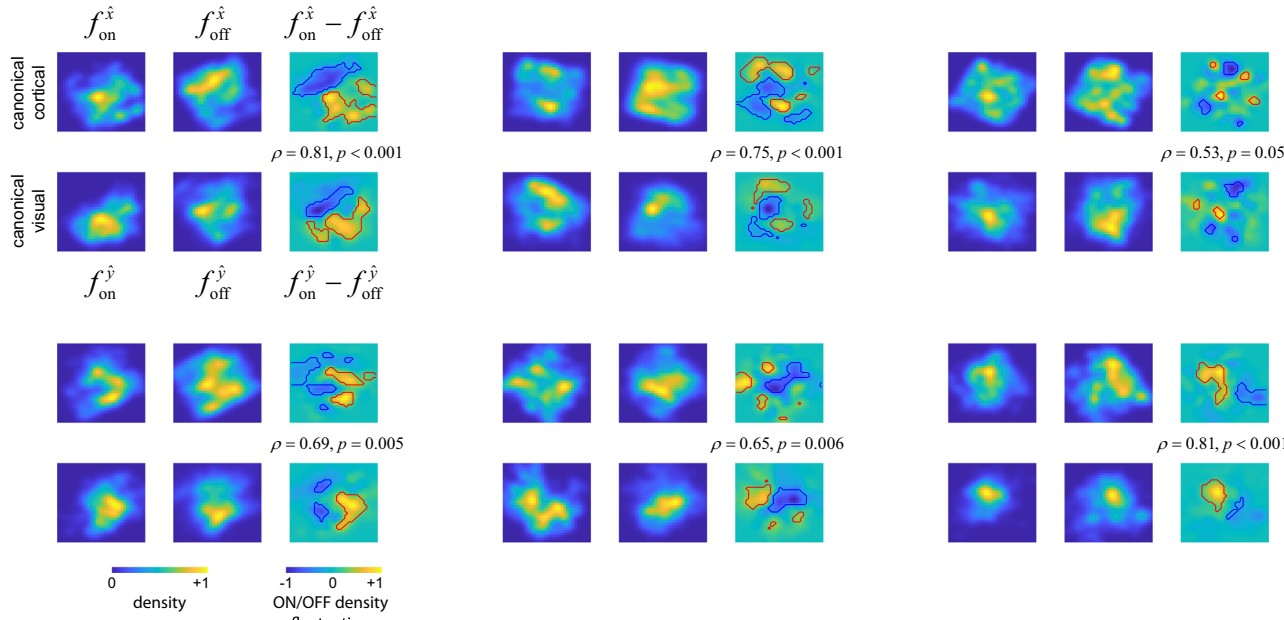

**Fig. 5 Correlation between ON/OFF domains and fluctuations in the balance of ON/OFF representation at the input.** Each panel displays the result of one experiment. In each case, the top row displays the density of ON ($f_{on}^{\hat{x}}$) and OFF ($f_{off}^{\hat{x}}$) cells in canonical cortical space, along with their difference, $f_{on}^{\hat{x}} - f_{off}^{\hat{x}}$. The bottom row shows the density in the position of receptive field centers for ON ($f_{on}^{\hat{y}}$) and OFF ($f_{off}^{\hat{y}}$) cells in canonical visual space, along with their difference, $f_{on}^{\hat{y}} - f_{off}^{\hat{y}}$. In all panels, both axes span the range from −2.5 to 2.5. Level sets showing areas where fluctuations are above or below the expected at the $p = 0.001$ significance level are shown by red and blue solid curves (Monte Carlo simulations with random shuffling of the ON/OFF labels in the population). The correlation coefficient between the fluctuations $f_{on}^{\hat{x}} - f_{off}^{\hat{x}}$ and $f_{on}^{\hat{y}} - f_{off}^{\hat{y}}$ is shown at the inset along with the statistical significance reached in each case (N = 1000 Monte Carlo simulations with reshuffling of ON/OFF labels). In each panel, the distributions are normalized by their maximum value and the colormap ranges from 0 to 1 (bottom left), while the differences of the distributions are shown normalized to their maximum absolute value, with the colormap ranging from −1 to 1.

bottom, solid dark lines). We conclude that the receptive field of simple cells can be explained by the non-negative, sampling of a handful of ON and OFF signals within a neighborhood of ~50 μm radius.

## Discussion

We measured the distributions of ON/OFF neurons across mouse V1 and the distribution of their receptive field centers across the visual field. We showed that, like other mammals[1–4], mouse V1 is parceled into ON/OFF domains. This provides an opportunity to study the contribution of ON/OFF maps to the cortical architecture in relative isolation, without the complexities that may arise from its interaction with ocular dominance and orientation maps[5]. Our findings revealed that fluctuations in the density of ON/OFF neurons on the cortical surface, which define ON/OFF domains, are mirrored by fluctuations in the density of ON/OFF receptive field centers on the visual field, as predicted by the biased-input model.

We further showed that the spatial distribution of ON/OFF domains appears to shape the receptive field structure of simple cells. First, in each cortical volume, the average, receptive field of simple cells correlates with the difference between the average of ON and OFF receptive fields (which we assume reflect approximate copies of the available afferent signals). Second, we demonstrated that the local diversity of simple-cell receptive fields can be explained by a model where neurons linearly combine a sparse number of ON and OFF signals within their cortical neighborhoods[16,22]. This result is consistent with a prior study by Smith and Häusser who established the sharing of ON/OFF signals by cortical cells[16]. Altogether, our findings suggest that ON/OFF domains originate in fluctuations of the spatial density of ON/OFF

inputs in the visual field which further shapes the two-dimensional structure of orientation tuned receptive fields[11–14,27,28,30].

Our results lend additional support to the proposal that ON/OFF domains are an important feature of thalamocortical connectivity that influences the architecture of the cortex[7,15], and are consistent with the idea that receptive fields in the cortex are constrained by the spatial distribution of ON and OFF inputs in the visual field[11,14,29,30]. In interpreting these data, we assumed that the receptive fields of mono-contrast ON and OFF neurons reflect those of geniculate afferents. We are currently testing this assumption in experiments where we image both the activity of thalamic boutons and cortical neurons. While the presence of ON/OFF domains may be required for the development of orientation tuning and two-dimensional, simple-cell structure, it is clearly not sufficient to establish robust orientation preference maps with the precision observed in higher mammals[18,36]. Instead, the development of orientation maps may additionally require a lower density of thalamic afferents, increased separation between ON and OFF domains, and lower retinotopic scatter of the thalamocortical projection than observed in mice[7,37]. Finally, it remains unknown if ON/OFF domains show global trends with retinotopy[38,39] or if they are related to other modular features of mouse V1, such as the system of M2 patches[19]—these are all-natural follow-up questions that require further research.

## Methods

**Experimental model and subject details.** All procedures were approved by UCLA's Office of Animal Research Oversight (the Institutional Animal Care and Use Committee) and were in accord with guidelines set by the U.S. National Institutes of Health. A total of six mice, male (2) and female (4), aged P35–56, were used. All these animals were from a cross between TRE-GCaMP6s line G6s2 (Jackson Lab, https://www.jax.org/strain/024742) and CaMKII-tTA (https://www.jax.org/strain/007004) where GCaMP6s is regulated by the tetracycline-responsive

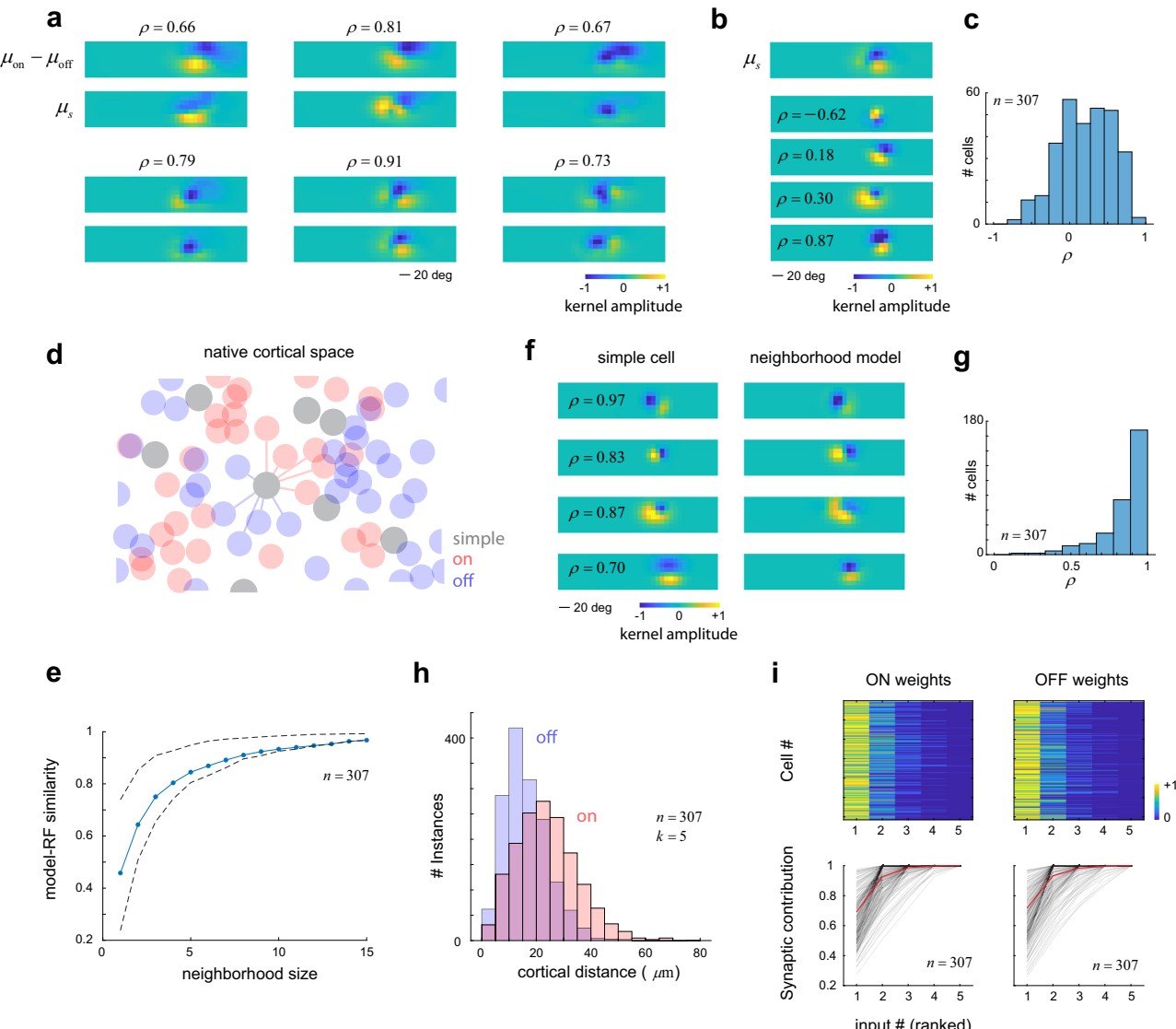

**Fig. 6 ON/OFF domains shape receptive field structure. a** Correlation between ON/OFF and simple-cell receptive fields. Each panel displays the result of one experiment. The top row displays the difference between the average ON ($\mu_{on}$) and OFF ($\mu_{off}$) receptive fields in the population, while the bottom row shows the average of all simple-cell receptive fields ($\mu_s$). The two are correlated ($p < 0.001$ in all cases). **b** Diversity of receptive fields in the population, showing the average along with the receptive fields of individual cells and their correlation coefficient with respect to the average. **c** Distribution of correlation coefficients between $\mu_s$ and those of individual cells in one experiment. **d** Modeling simple-cell receptive fields as linear combination of ON and OFF signals in a local neighborhood. The diagram shows a cartoon of a top view of a cortical patch of V1. ON and OFF cells define domains where one representation is dominant. Simple cells are assumed to pool ON and OFF signals within their neighborhood. **e** Model performance as a function of neighborhood size. Performance starts to saturate at a neighborhood size of $k$~5. Dashed lines indicate 25th and 75th percentiles of the distribution of correlation coefficients for each neighborhood size). **f** Example of four model fits. **g** Model performance for all neurons in one experiment for $k = 5$ showing the distribution of correlation coefficients between the actual receptive fields and their approximation by the model. **h** Distribution of distances for ON and OFF from simple cells for a neighborhood size of five—most cells are within $50\mu m$ of the target neuron. ON cells are statistically more distant than OFF cells ($p = 1.210^{-18}$, rank sum test). **i**,- Distribution of weights for ON and OFF neurons in decreasing order by rank (top). Weights are normalized by their total contribution. Two ON or OFF cells are sufficient to account for 90% of the total synaptic input to the neuron, as shown by the average cumulative distribution (bottom, red curve; individual cells shown by black curves). This dataset had $n_{on} = 379$, $n_{off} = 895$ and $n_{on+off} = 307$ cells (corresponding to dataset #7 in Table 1). Source data provided for panels **c**, **e**, **g**, **h**, **i**.

regulatory element (tetO). Mice were housed in groups of 2–3, in a protocol with reversed light cycle.

**Surgery**. All two-photon experiments were conducted through chronically implanted cranial windows over the primary visual cortex. Carprofen was administered pre-operatively (5 mg/kg, 0.2 mL after 1:100 dilution). Mice were anesthetized with isoflurane (4–5% induction; 1.5–2% surgery). Core body temperature was maintained at 37.5 °C using a feedback heating system. The eyes were coated with a thin layer of ophthalmic ointment to prevent desiccation. Anesthetized mice

were mounted in a stereotaxic apparatus. Blunt ear bars were placed in the external auditory meatus to immobilize the head. A portion of the scalp overlying the two hemispheres of the cortex (~8 mm by 6 mm) was then removed to expose the underlying skull. The skull was dried and covered by a thin layer of Vetbond. After the Vetbond dries (15 min) it provides a stable and solid surface to affix an aluminum bracket (a head holder) with dental acrylic. The bracket is then affixed to the skull and the margins are sealed with Vetbond and dental acrylic to prevent infections. We performed a craniotomy over monocular V1 on the left hemisphere using a high-speed dental drill. Special care was taken to ensure that the dura was not damaged during the process. Once the skull was removed, a sterile 3 mm

diameter cover glass was placed directly on the exposed dura and sealed to the surrounding skull with Vetbond. The remainder of the exposed skull and the margins of the cover glass were sealed with dental acrylic. Mice were then recovered on a heating pad. When alert, they were transferred back to their home cage. Carprofen was also administered post-operatively for 72 h. Mice were allowed to recover for at least 6 days before the first imaging session.

**Two-photon imaging**. We conducted imaging sessions in awake animals starting 6–8 days after surgery. Mice were positioned on a running wheel and head-restrained under a resonant, two-photon microscope (Neurolabware, Los Angeles, CA) controlled by Scanbox acquisition software and electronics (Scanbox, Los Angeles, CA). The light source was a Coherent Chameleon Ultra II laser (Coherent Inc, Santa Clara, CA). The excitation wavelength was set to 920 nm. The objective was an ×16 water immersion lens (Nikon, 0.8NA, 3 mm working distance). The microscope frame rate was 15.6 Hz (512 lines with a resonant mirror at 8 kHz). The field of view was $900 \times 540\,\mu m$ in all instances. The objective was tilted to be approximately normal the cortical surface. An electronically tuned lens (Optotune EL-10-30-C, Dietikon, Switzerland) was used to run independent sessions acquiring data from optical planes spaced $30\mu m$ apart starting at a depth of ~150 μm from the cortical surface. A total of 13 datasets were recorded each with a different number of optical sections (see table below). If we consider neurons at depths below 300 μm to be in layer 4[20], then we estimate about 75% of the data originate from layer 2 and 3, while 25% are from layer 4. Images were processed using a standard pipeline consisting of image stabilization, cell segmentation, and deconvolution using Suite2p (https://suite2p.readthedocs.io/). For anyone optical section, the location of the cells in the imaging plane was estimated as the center of mass of the corresponding region of interest calculated by Suite2p.

A camera synchronized to the frame rate of the microscope imaged the contralateral eye during data collection. These data were subsequently analyzed to determine the center and size of the pupil within the image plane. The distribution of eye movements was computed, yielding a mode and standard deviation. There were no obvious differences between the analyses performed on the entire dataset or on data segments where the eye position was restricted to lie within 1 SD of the mode. Here, we report the analysis using the entire dataset.

A summary of the datasets available is provided in the following table (asterisks indicate a significance level of 0.01):

**Visual stimulation**. We used a Samsung CHG90 monitor positioned 30 cm in front of the animal. The screen was calibrated using a Spectrascan PR-655 spectro-radiometer (Jadak, Syracuse, NY), generating gamma corrections for the red, green, and blue components via a GeForce RTX 2080 Ti graphics card. Visual stimuli were generated by a Processing sketch using OpenGL shaders (see http://processing.org). The screen was divided into an 18 by 8 grid (the average size of each tile was 8 by 8 deg of visual angle). Each frame of the stimulus was generated by selecting the luminance of each tile randomly as either bright (10% chance), dark (10% chance), or gray (80% chance). The stimulus was flashed for 166 ms and appeared at a rate of 1 per second. The screen was uniform gray between stimuli. Transistor-transistor logic (TTL) pulses generated by the stimulus computer signaled the onset of stimulation. These pulses were time-stamped by the microscope with the frame and line number that was being scanned at that moment the signals occurred. Sessions lasted for 25 min, generating the response of cells in the population to 1500 stimulus presentations. The expected number of bright or dark stimuli at any one location was 150 and its standard deviation was 11.6.

**Calculation of ON and OFF kernels**. For each cell and tile in the stimulus we calculated the average response the neuron locked to the presentation of a bright or dark patch over the 15 frames (1 sec) following stimulus onset. The ON kernel at a delay of $t$ frames after stimulus onset is represented as an image of equal size to the stimulus. The value of this image at tile location $(i, j)$ corresponds to the average response following the presentation of a bright stimulus at that location $t$ frames after stimulus onset. We denote this image by $ON(t)$ and adopt a similar definition of the OFF kernel, $OFF(t)$. For each time delay, we compute the norm of the kernel normalized by the norm at $t = 0$: $S_{on}(t) = ||ON(t)||/||ON(0)||$ and, similarly, we calculate $S_{off}(t) = ||OFF(t)||/||OFF(0)||$. These curves typically peak at delays of ~5 frames (corresponding to ~320 ms) (Fig. 2a). We declare a neuron to have a significant ON kernel if its normalized norm attained a peak value larger than 5 and a two-dimensional Gaussian fit of the kernel at the peak delay time accounts for at least 50% of the variance. A similar definition applies to OFF kernels. Thus, a neuron could have no significant maps, either significant ON or OFF maps (which we refer to as mono-contrast responses (Fig. 2b, top panel)), or both. The data from a neuron is included in our analyses if it has at least one significant kernel. The two-dimensional Gaussians fit ON and OFF kernels and yield their center locations $(y_1, y_2)$ on the visual field (Fig. 2b). For V1 cells with significant responses to both ON and OFF maps we compute the normalized distance between them, defined as the distance between the centers divided by the average standard deviation of the Gaussians. We define simple cells as those with a normalized distance larger than 0.5 (Fig. 2c)[24,25,40].

**Canonical correlation analysis**. Each neuron had assigned a coordinate in cortical space, $(x_1, x_2, x_3)$ (Fig. 1c) and, for each of its significant maps, one in visual space, $(y_1, y_2)$ (Fig. 2b, right panels). Canonical correlation analysis finds transformations $\hat{x} = A(x - \bar{x})$ and $\hat{y} = B(y - \bar{y})$ such that the covariance of $\hat{x}$ and $\hat{y}$ is diagonal and the correlations between matching canonical coordinates are maximized. The transformations are further constrained so that the variance of the canonical coordinates equals one. Note that in our case the matrix $A$ is $n \times 3$, while the matrix $B$ is $n \times 2$, where $n$ is the total number of cells with at least one significant map. We used MATLAB's `cannoncorr()` function for these analyses.

**Density estimation**. Given a distribution of points in native or canonical space (either cortical or visual) we estimate the density distribution by $f(x) = (1/n)\sum_{i=1}^{n} G_\sigma(x - x_i)$, where $G_\sigma(\cdot)$ is a two-dimensional Gaussian kernel of width $\sigma$ and $\{x_i\}$ $(i = 1, …, N)$ is the set of points under consideration[26]. For canonical variables, we chose a width of $\sigma = 0.25$, following the rule of thumb bandwidth estimator $0.9n^{-1/5}$, with $n$ ~500, which is a typical size for our data[26]. In native cortical space, we used $\sigma = 30\mu m$. Estimates of $f_{on}^x, f_{off}^x, f_{on}^y$ and $f_{off}^y$ and their counterparts in canonical space were all obtained by this procedure.

To evaluate the likelihood that the observed fluctuations could arise by chance, we randomly shuffled the labels of ON and OFF cells in $N = 1000$ experiments. For each experiment, we calculated the distribution of fluctuations at each point in cortical and visual spaces, which enabled us to compute $p = 0.001$ level sets (Fig. 2d, Supplementary Fig. 1, Fig. 5). Similarly, we computed the distribution of correlation coefficients between fluctuations of ON/OFF cells on the cortical surface and those of their central locations in the visual field, allowing us to calculate the statistical significance of the observed correlation in the original data (Fig. 5).

**Statistics and reproducibility**. We conducted experiments by independently measuring ON/OFF kernels in volumes of the primary visual cortex in 13 different instances (see Table 1). In each case, we covered a volume by measuring the

**Table 1 Summary of datasets.**

| Dataset # | mouse ID | sex | #planes | N_on | N_off | N_on + off | Simple | Complex | Layers correlation |
|---|---|---|---|---|---|---|---|---|---|
| 1 | P02 | F | 4 | 479 | 346 | 144 | 82 | 62 | 0.12 (**) |
| 2 | P02 | F | 7 | 821 | 936 | 415 | 248 | 167 | −0.01 (n.s.) |
| 3 | P03 | F | 6 | 274 | 294 | 47 | 33 | 14 | 0.29 (**) |
| 4 | P03 | F | 5 | 483 | 706 | 350 | 159 | 191 | 0.27 (**) |
| 5 | P04 | M | 8 | 319 | 549 | 95 | 64 | 31 | 0.15 (**) |
| 6 | P04 | M | 8 | 447 | 600 | 232 | 136 | 96 | 0.03 (**) |
| 7 | P03 | F | 6 | 379 | 895 | 307 | 131 | 176 | 0.31 (**) |
| 8 | P05 | F | 7 | 692 | 3037 | 1071 | 376 | 695 | 0.43 (**) |
| 9 | P05 | F | 8 | 383 | 1992 | 293 | 146 | 147 | 0.28 (**) |
| 10 | P06 | M | 7 | 317 | 1178 | 272 | 124 | 148 | 0.63 (**) |
| 11 | P06 | M | 8 | 310 | 524 | 62 | 27 | 35 | 0.55 (**) |
| 12 | P07 | F | 6 | 974 | 1263 | 419 | 163 | 256 | 0.41 (**) |
| 13 | P07 | F | 6 | 662 | 1052 | 723 | 316 | 407 | 0.17 (**) |

Each row in the table describes the attributes of one dataset. This includes the sex of the animal, the number of planes imaged, the number of mono-contrast ON and OFF cells, the number of cells with both ON and OFF kernels (along with the number of simple and complex cells within this category), and the correlation between $f_{on} - f_{off}$ computed in native cortical space from the top and bottom half of the optical sections recorded (asterisks indicate a significance level of 0.01).

responses of neurons in different optical sections. Each volume was collected in a single experimental session by recording each optical section in sequence. We did not estimate the number of neurons required for statistical significance of ON/OFF domains ahead of the experiments, as such number depends on the degree of spatial clustering expected (information that was not available before the study was conducted). Instead, we simply attempted to maximize the number of neurons recorded. As the study does not involve different groups undergoing different treatments, there was no need for randomization or blind assessment of outcomes. The statistical criteria used for data selection and to establish the significance of the results are described in the subsections above.

**Reporting summary**. Further information on research design is available in the Nature Research Reporting Summary linked to this article.

## Data availability

The data from all experimental sessions, including the processed kernels and estimated parameters, have been deposited at https://figshare.com/s/098b4ca29f5346648569. Source data are provided with this paper.

## Code availability

Sample code describing the structure of the database and the calculation of ON/OFF domains is provided together with the distribution of the data in the same Figshare repository.

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

## Acknowledgements

This work was supported by NIH grant NS116471 (D.L.R.).

## Author contributions

E.T. performed all surgeries. K.D. assisted with data analysis and collection. D.L.R. conceived the study, collected and analyzed the data, and wrote the manuscript.

## Competing interests

The authors declare no competing interests.
