## [Peer review file · Nature Communications]

REVIEWER COMMENTS

Reviewer #1 (Remarks to the Author):

This paper reports an important and timely discovery on the functional organization of ON and OFF pathways in mouse visual cortex. The authors convincingly demonstrate that ON and OFF thalamic afferents form ON and OFF cortical domains as previously demonstrated in other species. Moreover, they provide evidence that this ON-OFF segregation in cortical space is closely associated with a segregation of ON and OFF receptive fields in visual space, which provides the basis for cortical orientation selectivity. The paper is well written for a broad audience and should be of general interest. However, the canonical transformation in the data analysis needs to be explained in greater detail for both specialists and the general reader. This is important to interpret the results in terms of real cortical and visual space.

Major comments

1) The general reader needs to be provided with a more intuitive understanding of what the canonical space is and why the results cannot be demonstrated in real space. Perhaps, the reason is that the small cortical volume of mouse V1 and its sparse sampling of visual space introduce noise in the cortical representation and the canonical correlation analysis is needed to extract noisy information. Perhaps, the authors have other interpretation. However, they should make an effort at explaining in a bit more detail why a canonical transformation is needed to demonstrate ON and OFF domains in mice but not needed in other species with larger brains such as ferrets and cats.

2) It would be also helpful for the specialists to have access to a more detailed description of the canonical transformation of cortical and visual space. In a first read of the paper, it is difficult to understand how Figure 1f is calculated. The description in the main text is very brief and the figure legend does not add much additional information. The transformation would be much easier to understand if the main text and/or figure legend provided the canonical variables or linear equations used in the transformation. For example, the figure legend could include a line as this: 'In this specific example, $\hat{x}_1 = a_1 x_1 + b_1 x_2 + c_1 x_3$, $\hat{x}_2 = a_2 x_1 + b_2 x_2 + c_2 x_3$, $\hat{y}_1 = a_3 y_1 + b_3 y_2$, $\hat{y}_2 = a_4 y_1 + b_4 y_2$ '. Then, in the methods section, the authors could provide the average and standard deviation of the canonical variables (a_{1-4} , b_{1-4} , c_{1-2}). This will give the reader a better understanding of what exactly the transformation is doing. For example, are the canonical variables smaller for cortical depth (x_3) than other dimensions because retinotopy changes more with x_1 and x_2 than x_3 ?

3) Some additional explanation in the figure legends would also help. For example, how are the x_1 , x_2 , x_3 dimensions of cortical space plotted in two dimensions in Figure 1e? Is the depth dimension

projected in the same two dimensional plane making each point in Figure 1e being plotted based only on its x1, x2 values? A brief sentence in the figure legend stating this would be very helpful.

4) There seems to be a typo in the explanation provided in methods, which also adds to the confusion. The methods indicate that: 'Canonical correlation analysis finds transformations $\hat{x} = A(x - \bar{x})$ and $\hat{y} = A(y - \bar{y})$ such that the covariance of \hat{x} and \hat{y} is diagonal and the correlations between matching canonical coordinates are maximized.'

Is $\hat{x} = A(x - \bar{x})$ and $\hat{y} = A(y - \bar{y})$

or

$\hat{x} = A(x - \bar{x})$ and $\hat{y} = B(y - \bar{y})$?

Is the analysis done using something equivalent to $[A,B,r,U,V] = \text{canoncorr}(X,Y)$ in Matlab, where X is $[x1, x2, x3]$ and Y is $[y1, y2]$? If this is the case, reporting the Matlab/Python function or providing the code would also help to understand better the analysis.

5) Is it possible to demonstrate some form of columnar organization of these ON and OFF domains in mice? For example, is it possible to perform canonical correlations for x1, x2 and then show that the ON and OFF domains found in cortical space are preserved through the cortical depth (x3)?

6) Whereas the authors make clear that the ON and OFF domains cannot be demonstrated without canonical correlation, it would be helpful to find a way to estimate the average size of the domains in real cortical space. This is important to compare the results obtained in mice with other species. Also, are the measurements consistent with previous estimates of ON and OFF domains in mice from Smith and Hausser (2010)? Is it possible to demonstrate ON-OFF cortical clustering in the authors' data using the analysis from Smith and Hausser (2010)?

7) The section discussing three possible models is difficult to understand. It would be helpful to expand the explanation and the arguments against the models not favored by the authors. Specifically, this sentence is difficult to unpack: 'The specific convergence model would not yield the observed correlations in relative density fluctuation and can be rejected'.

Minor

- 1) Please, indicate the approximate eccentricity of the recordings or simply state whether the recordings were done in monocular cortex, binocular cortex or both. This is important to compare the results with other species that have more extensive binocular cortices.
- 2) Please, explain the color code in Figure 1c. The colors of the traces at the bottom illustrate different cell examples but what are the colors at the top? What do green, purple, yellow, orange and blue represent? What do the cells with the same color have in common?
- 3) Typo in page 6. 'Instead, orientation columns are maps may additionally require a lower density of thalamic afferents...' should be 'Instead, orientation maps may additionally require a lower density of thalamic afferents...'.
- 4) The authors divide receptive fields in ON, OFF, simple and complex, but we do not learn much about the proportion and distribution of these cells in cortex. What percentage of cells are in each of these four categories? How are the four cells distributed in cortical space? Are ON and OFF cells more likely to be deeper in the cortex than simple and complex cells?
- 5) Is figure 3d real cortical space or canonical cortical space? If it is canonical cortical space, please change the label in the figure to avoid confusion. It would be also helpful to have a scale bar. Is this figure representing real or simulated data? The reader will probably assume that is real data but this should be stated somewhere in the figure legend.
- 6) It would be helpful to report whether there is a significant difference between the ON-ON and OFF-OFF cortical distance (Figure 3h) and between the ON and OFF synaptic weights (Figure 3i) estimated from the fitting analysis. It seems that there is a tendency for greater scatter in ON than OFF receptive fields as demonstrated in cats, tree shrews, and mice (by Ringach's team). It would be nice to state this somewhere in the paper.
- 7) It would be really exciting if the authors could take their study one step forward and show that the orientation predicted from the neighborhood model matches the measured orientation (even if it is a small bias). The authors can ignore this comment if they do not have measures of orientation preference.

Reviewer #2 (Remarks to the Author):

ON/OFF domains shape receptive fields in mouse visual cortex

Elaine Tring, Konnie Duan, Dario L. Ringach

Tring and colleagues use two-photon calcium imaging to answer the question whether mouse primary visual cortex is parceled into light on and light off responsive domains. After aligning visual and cortical space with canonical correlation analysis they find that there are segregated ON and OFF regions. They go on to show that, for a given cortical region, the average simple cell receptive field (RF) correlates with the difference between the average ON and OFF RFs. Finally, using a model, they demonstrate that simple cell RFs can be constructed from a small number of local ON and OFF inputs.

The segregated termination of ON and OFF inputs has been observed in in the visual cortex of a number of higher mammals, and might be related to the systematic mapping of orientation preference in these species. Mice lack such an organization for preferred orientation, thus the observation of ON and OFF domains in this species is very relevant, since it would indicate that there is no strict coupling between ON and OFF domains and the orientation map. Moreover, ON/OFF domains would be the first ever described orderly mapped response property in mouse visual cortex, apart from the retinotopic map.

In its present state, however, the manuscript falls short of convincingly demonstrating ON and OFF domains in mouse visual cortex.

1.

The paper is missing basic methodological information, crucial for proper scientific review. The authors need to provide information on these points:

- Did the surgery include a craniotomy with a cranial window implant?
- Please provide more details on Carprofen analgesia.
- Was two-photon imaging performed under anesthesia or awake?
- Were eye-movements tracked and compensated for during two-photon imaging?
- Please define exactly what is meant by “datasets/experiments”. How many imaging planes were acquired per mouse?

Apart from these specific points, the authors should carefully check their methods to make sure that all relevant information is provided.

2.

The authors acquired a 540 x 900 μm^2 FOV in mouse V1. However, it is unclear at which relative and absolute retinotopic position each FOV was taken. As higher visual areas could readily affect RF clustering, we ask the authors to show the V1 borders within each FOV, e.g. measured via intrinsic signal imaging or inferred from the mapped retinotopy. Related, we ask the authors to add exact axis tick labels for visual space (e.g. Fig 1e), in order to provide insight into the absolute retinotopy/visuotopy of the extracted RFs.

3.

The authors use an uncommon way of generating sparse noise visual stimuli for RF mapping, which diverges from established sparse noise stimuli. These latter stimuli include a local exclusion criterion

within and between subsequent stimulus-frames to prevent, for example, the creation of oriented edges by neighboring patches, and consequently ensure purely spatially driven RF responses. This was, however, not implemented in this current study. We are especially concerned about oriented edges within a given stimulus-frame, as in the example in Fig 1a, which will elicit orientation-driven responses, rather than purely spatial RF responses. We therefore ask the authors to provide the following information:

- Was the patch coverage randomized per stimulus frame or over all 1500 stimulus frames? In other words, was each patch in the visual field covered with the same number of repetitions?
- Are the 1500 stimulus frames sufficient to average out any orientation effects? This could be tested by comparing RFs based on subsampling from the 1500 stimuli.
- Please provide example RF calcium response transients (trial and average) to verify the accuracy of this novel RF stimulation.

4.

Related, each stimulus frame was only presented for 166ms, however, the authors consider multiple successive inter-stimulus interval frames for determining RFs. While it is reasonable to integrate over several frames due to the slow calcium kinetics of GCaMP6s, we are concerned, that this approach won't let the authors distinguish between ON and OFF subfields. The considered calcium response transient includes both the stimulus onset and offset back to the grey background. For example, for a white patch at a given location in visual space, the calcium transient would represent an integrated signal of both, the stimulus onset (in this case light increment) and stimulus offset (a light decrement). Given the slow kinetics of GCaMP6s, ON responses can therefore not be precisely distinguished from OFF responses. This has major consequences on the interpretability of the results of clustering of ON and OFF RFs and could very well be a reason for overlapping ON and OFF RF subfields, defined as "complex cells" by the authors. We suggest to instead present each stimulus frame for a longer duration and only take the frames during stimulation into account for RF extraction, in order to ensure precise extraction of the RF subfield polarity/sign.

5.

To reveal ON/OFF domains, the authors employed a canonical correlation analysis (CCA), linking visual and cortical space. They state "It was convenient, for reasons that will become clear soon,...", but it does not become clear why this analysis was needed. They should elaborate on the use of CCA. For example, what does it mean if clustering is observed in a space that was optimized for maximum correlation? Please provide intuition on what canonical cortical/visual space is/means. Also, please show CCA weight distributions for each vector.

Why was CCA used in the first place? Were the actual domains in cortical space to diffuse or noisy to become clearly visible? Was there too much variance between the few (6) mice imaged to draw firm conclusions? Ideally, one would like to actually see the domains in the cortex. For this, it might be useful to focus on layer 4, since this is where one would expect the clearest segregation. The paper does not even attempt to analyze the layer specific organization of ON/OFF domains. Instead, it is stated that "The inclusion of cortical depth (x3) allowed us to compensate for slight departures of the objective from the surface normal", but it is not explained how this is done.

6.

The maps shown in Fig.1 e, f contain large regions with only OFF responses. What is the explanation for this large-scale uneven distribution? Further, there are gaps in the maps, probably due to blood vessels. How do these interfere with the detection of ON/OFF clusters? What are the actual numbers of ON and OFF RFs?

7.

The arguments on the distinction between different models of how ON/OFF domains are generated (Fig.2 b-d) are hard to follow. In particular, are the authors confident that given the rather indirect determination of neuronal activity (GCaMP6s, then spike inference), they can exclude a contribution of “synaptic modulation”? The latter phrase is also misunderstandable; I guess “synaptic strengths” captures it better.

8.

Some of the figure plots miss crucial information; please check for correct and informative axis labels, and for scale bars. It is very unclear what the color code in Fig.1 c is.

9.

The authors state that “ON/OFF domains are a critical feature of thalamocortical connectivity”. While this is certainly true for some species, they do not actually show this for the mouse, since they do not record from input fibers, and since it is not clear to which degree their findings hold for layer 4, where thalamic axons mainly terminate,

10.

The very relevant paper by Smith and Häusser is cited, but just in passing. It should be given credit in the conclusions, too.

Reviewer #3 (Remarks to the Author):

It is thought that On/Off domains of receptive field in primary visual cortex (V1) are derived from thalamocortical afferents, which provide for the columnar cortical architecture of orientation maps observed in carnivores and primates. The paper by Tring, Duan and Ringach shows that On and Off receptive field domains in layer 2/3 of mouse V1 form distinct 100 um-wide clusters. Spatial clustering is an unexpected feature of mouse V1, whose orientation tuned neurons are randomly distributed like salt-and-pepper. The newly found order is intriguing and adds to the emerging evidence of a modular architecture of mouse V1.

1) While the evidence for spatial clustering is quite convincing. It is important to show the distribution of these clusters across V1. Figures 1e and f take a stab at this, but it is difficult to extract specific information from the illustrations.

2) Figure 1d and 2a show that the receptive field of a single simple cell representing a given point in visual space is made-up from a pair of On and Off subfields. The authors may consider extending the analysis to the cortical point image, which would give a more complete picture of the number and distribution of On and Off subfields and their possible relationship to previously reported spatially clustered patterns in mouse V1.

3) The authors reason that the spatial clustering of On and Off subfields may be related to orientation selectivity. I have difficulties to understand how the apparent order found in their study squares with the disorderly distribution of orientation tuned neurons in mouse V1.

4) In Figure 1d, the panels are too small, making it difficult to extract RF size and spatial displacement of ON and Off subregions. Please provide scale for deg of visual field. How large are the receptive fields of simple and complex cells?

5) In Figure 1e, how many On and Off neurons/10 deg?

6) I recommend using consistent language for domain, subregion, and On/Off maps throughout the text.

7) In Figure 1g, there is a discrepancy of correlation of the second co-variances $p=0.76$ (text) and $p=0.81$ (Figure 1g).

8) In Figures 1f, g, Please provide scales for cortical distance (μm) and space (deg). It is difficult to extract the center-to-center distances between On and Off cells, and to get a sense how many On and off subfields are contained within a cortical region representing a specific location in space.

9) In Figures 1e, f, please specify in which part of the visual field the recordings were taken and provide axes for elevation and azimuth.

10) In Figure 2a, the title should indicate that these are probability distributions. Maybe I do not read the plots correctly, but it would be helpful to add scales for distance and visual angle.

11) Page 4/para 3/line 2. Shouldn't it read On/Off domain (subregions) centers instead of receptive field centers?

12) In Figures 3a, b, f, please provide scales for cortical distance and visual space.

13) In the legend of Figure 3e please replace by "3" by "e".

14) Page 6/para 1, I suggest to improve the last sentence. Clearly indicate that the receptive field is composed of 100 um-wide On and Off clusters.

15) Page 6/para 4, please revise 2nd to last sentence.

We would like to thank the reviewers for their thorough and constructive comments. We have revised the manuscript to address their concerns and questions. We identified two major issues evident in the reviewer's comments. First, we realized from the questions raised that there was a need for a clearer description of what canonical correlation analysis is trying to achieve, provide an intuition behind the analysis, and explain to what extent the results depend on using such method. Second, we performed additional analyses to address reviewer #2 technical concerns regarding the use of our sparse noise stimuli and the analysis of these data. The reviewers also offered various suggestions for improvement, nearly all of which were adopted. We hope the revised manuscript and the point-by-point reply to the reviewer's comments provide an adequate response to their concerns and questions.

All changes to the manuscript have been tracked. We are also submitting a clean version of the text, as the number of changes is extensive and makes reading the marked-up manuscript nearly impossible.

REVIEWER COMMENTS

Reviewer #1 (Remarks to the Author):

This paper reports an important and timely discovery on the functional organization of ON and OFF pathways in mouse visual cortex. The authors convincingly demonstrate that ON and OFF thalamic afferents form ON and OFF cortical domains as previously demonstrated in other species. Moreover, they provide evidence that this ON-OFF segregation in cortical space is closely associated with a segregation of ON and OFF receptive fields in visual space, which provides the basis for cortical orientation selectivity. The paper is well written for a broad audience and should be of general interest. However, the canonical transformation in the data analysis needs to be explained in greater detail for both specialists and the general reader. This is important to interpret the results in terms of real cortical and visual space.

Major comments

1) The general reader needs to be provided with a more intuitive understanding of what the canonical space is and why the results cannot be demonstrated in real space. Perhaps, the reason is that the small cortical volume of mouse V1 and its sparse sampling of visual space introduce noise in the cortical representation and the canonical correlation analysis is needed to extract noisy information. Perhaps, the authors have other interpretation. However, they should make an effort at explaining in a bit more detail why a canonical transformation is needed to demonstrate ON and OFF domains in mice but not needed in other species with larger brains such as ferrets and cats.

We can see how our terse description caused some confusion. To clarify, canonical correlation analysis is **not** required to demonstrate the presence of ON/OFF domains. ON/OFF domains can readily be demonstrated by measuring variations in the relative density of ON or OFF neurons across the cortical surface in the **native** coordinate system (the imaging plane), as has been done in other species. The significance of the fluctuations can be assessed by random relabeling of the cells (by shuffling their ON and OFF labels). The three examples below, show ON/OFF domains in three experiments computed in native cortical space. The contours show level sets at a significance of 0.01.

It is important to realize that the structure of ON/OFF domains observed in native cortical space is also going to be reflected in canonical cortical space, which is nothing more than a linear transformation of the images above. To make this point clear we have added labeled points to **Fig 1e,f** and discuss in the text how the ON/OFF domains observed in native space (such as a dominance of OFF cells at one location) will also be observed in the canonical space.

If ON/OFF domains can be shown in either native or canonical cortical space, why do we opt to present the results in canonical space? If all we wanted to do is show the existence of OFF and ON domains on the cortex, we could do it directly on cortical space, as shown by the figure above. However, an important part of the analysis investigates if the relative fluctuation in the density of ON and OFF cells in the cortex is related to the distribution of the centers of ON and OFF receptive fields in the visual field. The answer to this question **requires** the alignment of cortical and visual space. Canonical correlation is one way to achieve this which has some nice properties, such as yielding orthogonality between the axes and treating both cortical and visual space symmetrically, but it is not certainly the only possible approach. An alternative, for example, would be to map the location of the receptive fields onto the cortex by using the best linear approximation of the retinotopic map.

2) It would be also helpful for the specialists to have access to a more detailed description of the canonical transformation of cortical and visual space. In a first read of the paper, it is difficult to understand how Figure 1f is calculated. The description in the main text is very brief and the figure legend does not add much additional information. The transformation would be much easier to understand if the main text

and/or figure legend provided the canonical variables or linear equations used in the transformation. For example, the figure legend could include a line as this: 'In this specific example, $\hat{x}^1 = a_1 x_1 + b_1 x_2 + c_1 x_3$, $\hat{x}^2 = a_2 x_1 + b_2 x_2 + c_2 x_3$, $\hat{y}^1 = a_3 y_1 + b_3 y_2$, $\hat{y}^2 = a_4 y_1 + b_4 y_2$ '. Then, in the methods section, the authors could provide the average and standard deviation of the canonical variables (a_{1-4} , b_{1-4} , c_{1-2}). This will give the reader a better understanding of what exactly the transformation is doing. For example, are the canonical variables smaller for cortical depth (x_3) than other dimensions because retinotopy changes more with x_1 and x_2 than x_3 ?

The basic description above is correct. Canonical correlation is nothing more than a linear transformation of the data which also satisfies orthogonality constraints and normalizes the data to have unit variance. Following the reviewer's suggestion, we now include a more intuitive description of canonical correlation analysis in the text.

3) Some additional explanation in the figure legends would also help. For example, how are the x_1 , x_2 , x_3 dimensions of cortical space plotted in two dimensions in Figure 1e? Is the depth dimension projected in the same two dimensional plane making each point in Figure 1e being plotted based only on its x_1 , x_2 values? A brief sentence in the figure legend stating this would be very helpful.

We have added additional explanations to the figure legends as suggested by the reviewer. Briefly, x_3 is coming out of the page in **Fig 1e** (now stated in the legend), so yes – we are basically looking at a projection into the (x_1 , x_2) plane in this figure.

4) There seems to be a typo in the explanation provided in methods, which also adds to the confusion. The methods indicate that: 'Canonical correlation analysis finds transformations $\hat{x} = A(x - \bar{x})$ and $\hat{y} = A(y - \bar{y})$ such that the covariance of \hat{x} and \hat{y} is diagonal and the correlations between matching canonical coordinates are maximized.'

Is $\hat{x} = A(x - \bar{x})$ and $\hat{y} = A(y - \bar{y})$

or

$\hat{x} = A(x - \bar{x})$ and $\hat{y} = B(y - \bar{y})$?

Is the analysis done using something equivalent to $[A,B,r,U,V] = \text{canoncorr}(X,Y)$ in Matlab, where X is [x_1 , x_2 , x_3] and Y is [y_1 , y_2]? If this is the case, reporting the Matlab/Python function or providing the code would also help to understand better the analysis.

Thank you -- the typo has been fixed. Indeed, these analyses were performed using MATLAB's `cannoncorr()`. This is now stated under **Methods**. We plan to post the data and code in Figshare if the paper is accepted for publication.

5) Is it possible to demonstrate some form of columnar organization of these ON and OFF domains in mice? For example, is it possible to perform canonical correlations

for x_1 , x_2 and then show that the ON and OFF domains found in cortical space are preserved through the cortical depth (x_3)?

The methods we use pool the data across the cortical depth to increase our statistical power and demonstrate the clustering of ON/OFF responses. Note that if we did not have consistency across depth, pooling across depth would not yield increasingly significant results. Thus, in a sense, the fact that our methods work is consistent with the notion that the maps are columnar. Unfortunately, it very difficult to obtain sufficient data from individual optical slices to show the organization at < 0.01 statistical level which would allow the comparison of ON/OFF maps across different depths as the reviewer suggested.

6) Whereas the authors make clear that the ON and OFF domains cannot be demonstrated without canonical correlation, it would be helpful to find a way to estimate the average size of the domains in real cortical space. This is important to compare the results obtained in mice with other species. Also, are the measurements consistent with previous estimates of ON and OFF domains in mice from Smith and Hausser (2010)? Is it possible to demonstrate ON-OFF cortical clustering in the authors' data using the analysis from Smith and Hausser (2010)?

We apologize once again for causing a confusion about this issue. As we explain in point #1 above (and in the revised text as well), ON/OFF domains can certainly be demonstrated **without** the use of canonical correlation. We also added the average size of ON/OFF domains in the text as requested.

Our findings are consistent with the previous work of Smith and Hausser in that there is agreement that the receptive field a cortical population appear to be assembled by sampling from a small set of ON and OFF signals within their local neighborhood. We have now added this the discussion. Note this prior work did not show the large-scale organization of ON and OFF domains as demonstrated in our study.

7) The section discussing three possible models is difficult to understand. It would be helpful to expand the explanation and the arguments against the models not favored by the authors. Specifically, this sentence is difficult to unpack: 'The specific convergence model would not yield the observed correlations in relative density fluctuation and can be rejected'.

Thank you. We have rewritten the discussion of the models and hopefully the claims are now clear.

Minor

1) Please, indicate the approximate eccentricity of the recordings or simply state

whether the recordings were done in monocular cortex, binocular cortex or both. This is important to compare the results with other species that have more extensive binocular cortices.

All recordings are from monocular cortex. We have now added the eccentricity information in the Methods section. See also our reply to Reviewer #2, point #2 below, who requested similar information.

2) Please, explain the color code in Figure 1c. The colors of the traces at the bottom illustrate different cell examples but what are the colors at the top? What do green, purple, yellow, orange and blue represent? What do the cells with the same color have in common?

Different ROIs are shown in a random color, so they are discriminable from each other. We have now stated this in the figure caption.

3) Typo in page 6. 'Instead, orientation columns are maps may additionally require a lower density of thalamic afferents...' should be 'Instead, orientation maps may additionally require a lower density of thalamic afferents...'.
Thank you. Fixed.

4) The authors divide receptive fields in ON, OFF, simple and complex, but we do not learn much about the proportion and distribution of these cells in cortex. What percentage of cells are in each of these four categories? How are the four cells distributed in cortical space? Are ON and OFF cells more likely to be deeper in the cortex than simple and complex cells?

We now provide numbers for ON, OFF, cells with at least two subregions of opposite sign, which we term ON+OFF, and their division into simple and complex cells (see new table in Methods)

For the ON+OFF population of cells, the normalized distance has the distribution shown on the right. We defined simple cells as those with a normalized distance larger than 0.5. For a normalized distance of 0.5, the distance between ON and OFF subregions is such that we often see a residual, linear kernel for cells with responses to both polarities, which is what we model in Fig 4.

The question of the spatial distribution of cells is indeed an important one. We have some preliminary analyses supporting the notion that simple cells tend to be located between ON/OFF domain peaks. This seems consistent with what is observed in long horizontal penetrations in the cat, where cells with both ON and OFF subregions are found between groups of neurons responding mostly to one polarity. We feel a full description of these data is outside the scope of this paper. We plan to report on these results separately.

5) Is figure 3d real cortical space or canonical cortical space? If it is canonical cortical space, please change the label in the figure to avoid confusion. It would be also helpful to have a scale bar. Is this figure representing real or simulated data? The reader will probably assume that is real data but this should be stated somewhere in the figure legend.

Figure 3d is native cortical space and the illustration is just a cartoon to explain the structure of the model. We have clarified this in the text and figure caption.

6) It would be helpful to report whether there is a significant difference between the ON-ON and OFF-OFF cortical distance (Figure 3h) and between the ON and OFF synaptic weights (Figure 3i) estimated from the fitting analysis. It seems that there is a tendency for greater scatter in ON than OFF receptive fields as demonstrated in cats, tree shrews, and mice (by Ringach's team). It would be nice to state this somewhere in the paper.

Yes, that difference between the distances is significant (now stated in the text). But this directly reflects the fact that OFF neurons are more numerous (see table in Methods) and does not really speak directly as to the RF scatter.

A higher scatter of ON cells compared to OFF cells can be revealed in the present dataset if we perform the canonical correlation analysis for ON and OFF cells *separately* and then comparing the correlations obtained (**Fig 1g**).

The higher the correlation between visual and cortical canonical variables, the smaller the scatter. When we do this analysis (figure above), we clearly observe that OFF cells have lower scatter than ON cells, as the correlations between the first (blue) and second canonical variables (red) are higher for OFF than ON cells. The results are significant ($p < 0.001$, paired rank-sum test). We now state this result in the text.

7) It would be really exciting if the authors could take their study one step forward and show that the orientation predicted from the neighborhood model matches the measured orientation (even if it is a small bias). The authors can ignore this comment if they do not have measures of orientation preference.

We think the reviewer is asking for an independent assessment of orientation tuning using, for example, drifting gratings. Unfortunately, we did not collect such data in these long experiments where we tried to maximize the size of our population by measuring as many optical planes as possible.

Given what is known, we would expect such correlations to be present, as multiple, prior findings demonstrate a tight relationship between the axis of displacement of ON and OFF subregions and the preferred orientation of neurons. However, we do not have such data to offer a direct comparison here.

Reviewer #2 (Remarks to the Author):

ON/OFF domains shape receptive fields in mouse visual cortex

Elaine Tring, Konnie Duan, Dario L. Ringach

Tring and colleagues use two-photon calcium imaging to answer the question whether mouse primary visual cortex is parceled into light on and light off responsive domains. After aligning visual and cortical space with canonical correlation analysis they find that there are segregated ON and OFF regions. They go on to show that, for a given cortical region, the average simple cell receptive field (RF) correlates with the difference between the average ON and OFF RFs. Finally, using a model, they demonstrate that simple cell RFs can be constructed from a small number of local ON and OFF inputs.

The segregated termination of ON and OFF inputs has been observed in in the visual cortex of a number of higher mammals, and might be related to the systematic mapping of orientation preference in these species. Mice lack such an organization for preferred orientation, thus the observation of ON and OFF domains in this species is very relevant, since it would indicate that there is no strict coupling between ON and OFF domains and the orientation map. Moreover, ON/OFF domains would be the first ever described orderly mapped response property in mouse visual cortex, apart from the retinotopic map.

In its present state, however, the manuscript falls short of convincingly demonstrating ON and OFF domains in mouse visual cortex.

1.

The paper is missing basic methodological information, crucial for proper scientific review. The authors need to provide information on these points:

- Did the surgery include a craniotomy with a cranial window implant?

We have added a description of the craniotomy to the surgery sub-section in the **Methods**.

- Please provide more details on Carprofen analgesia.

Added to the **Methods**.

- Was two-photon imaging performed under anesthesia or awake?

We apologize for this omission. Yes, the animals were awake and free to walk on a wheel. We have stated this now under the “two-photon imaging” sub-section of the method and the first sentence of the results.

- Were eye-movements tracked and compensated for during two-photon imaging?

Indeed. We added a paragraph to the “two-photon imaging” sub-section on how we used these data. Note that eye movements can only work against the relationships we report (as they will add noise to the retinotopic locations of the receptive fields) – they will not be able to create them.

- Please define exactly what is meant by “datasets/experiments”. How many imaging planes were acquired per mouse?

A dataset consists of a series of consecutive, sparse-noise experiments at different optical planes in one experimental session. We have now added a table in the methods that describes each dataset, including the sex, number of planes and on/off kernels measured.

Apart from these specific points, the authors should carefully check their methods to make sure that all relevant information is provided.

We believe all the information required to replicate the results has been included. We would be happy to provide any additional information the reviewer deems important.

2. The authors acquired a 540 x 900 μm^2 FOV in mouse V1. However, it is unclear at which relative and absolute retinotopic position each FOV was taken. As higher visual areas could readily affect RF clustering, we ask the authors to show the V1 borders within each FOV, e.g. measured via intrinsic signal imaging or inferred from the mapped retinotopy. Related, we ask the authors to add exact axis tick labels for visual space (e.g. Fig 1e), in order to provide insight into the absolute retinotopy/visuotopy of the extracted RFs.

We routinely measure a coarse retinotopy of our field before we begin each session. We do this by splitting the imaging plane into a 3x3 grid (as shown in the figure below) and quickly mapping the retinotopy of these signals. The location at which we get a peak response for each signal is plotted by a corresponding number in the visual field along with the average responses for the entire field. We know we are recording in V1 because (a) we center of our craniotomies at the average location of V1 based on prior studies, (b) we obtain a lawful retinotopic map without reversals (which we would obtain if the imaging field were to impinge into adjacent, higher visual areas), (b) the sign of the map is -1 (the map is mirror reversed). This coarse retinotopy also allow us to estimate the center of aggregate receptive field of the population, which was 24 deg in azimuth and 5 deg elevation, in our experiments. We have now included a brief description of this methodology in the Methods section.

3.

The authors use an uncommon way of generating sparse noise visual stimuli for RF mapping, which diverges from established sparse noise stimuli. These latter stimuli include a local exclusion criterion within and between subsequent stimulus-frames to prevent, for example, the creation of oriented edges by neighboring patches, and consequently ensure purely spatially driven RF responses. This was, however, not implemented in this current study. We are especially concerned about oriented edges within a given stimulus-frame, as in the example in Fig 1a, which will elicit orientation-driven responses, rather than purely spatial RF responses. We therefore ask the authors to provide the following information:

The calculation of the linear kernel of a linear-nonlinear system by means of cross-correlation requires the input to be white. Our stimulus is indeed spatially white and has been used previously to measure the linear component of receptive fields.

The stimulus the reviewer is probably referring is one employed by our colleagues at the Allen institute (https://observatory.brain-map.org/visualcoding/stimulus/locally_sparse_noise), who opted to create such an exclusion zone around checks that are bright or dark.

It is not entirely clear why this choice was made and, as far as we can tell, the original manuscript does not provide a reason. We conjecture they did so to counteract the effects of surround suppression in mouse V1, which can be substantial. Incidentally,

this is also the reason why we chose to have a sparse stimulus, where only 20% of the checkers are either dark or bright in any one frame (as opposed to dense, white noise).

Although in our stimuli dark and bright stimuli they can certainly appear at adjacent locations, this is not expected to have an effect in the estimate of the first order kernel, which only requires the input to be spatially white. Nevertheless, as suggested by the reviewer we carried out the control analyses where we performed kernel estimation based on a subset of our stimuli that contained exclusion zones around each location and observed no obvious differences (see figure below).

- Was the patch coverage randomized per stimulus frame or over all 1500 stimulus frames? In other words, was each patch in the visual field covered with the same number of repetitions?

Each frame was generated randomly by the procedure described. So, each location has a variable number of repetitions. This is of course taken into consideration when computing the **mean** response to each stimulus polarity and location.

- Are the 1500 stimulus frames sufficient to average out any orientation effects? This could be tested by comparing RFs based on subsampling from the 1500 stimuli.

Following the suggestion of the reviewer, we recalculated the kernels by selecting sub-sets of the stimuli that, by chance, had an exclusion zone around a location.

This was done as follows. For each location, we selected the subset of stimuli where a bright checker appeared with a 3x3 exclusion zone (that is, none of its neighbors had a stimulus). We used these images to compute the responses of neurons to the ON stimulus at that location. We then selected a subset of stimuli where a dark checker appeared at that location with a 3x3 exclusion zone. WE used those images to compute the responses of neurons to the OFF stimulus at that location. The same was done for each location and the results compiled into ON/OFF kernels.

The image on the right shows the raw kernels of 4 neurons estimated with the original sparse stimulus (left column) and using a subset of images where, at each location, the stimuli were presented in isolation (right column). There are no obvious differences between these results.

- Please provide example RF calcium response transients (trial and average) to verify the accuracy of this novel RF stimulation.

The figure on the right shows the stimulus-triggered fluorescence and spike inference signals (both z-scored). Recall that the stimulus is presented at a rate of 1 Hz, which corresponds to 15.5 frames of the microscope, so here we see slightly over a cycle of the fluorescence signal. We see the fluorescence signal showing a single, early peak and a slower decay. The spike inference signal shows a sharper early peak which coincides with the rising phase of the calcium signal as expected (as the deconvolution algorithm basically detects sharp, positive deflections in fluorescence). These signals show the responses are modulated at the temporal frequency of the stimulus. The kernels show a similar temporal dependence (see reply to the reviewer's next point).

4.

Related, each stimulus frame was only presented for 166ms, however, the authors consider multiple successive inter-stimulus interval frames for determining RFs. While it is reasonable to integrate over several frames due to the slow calcium kinetics of GCaMP6s, we are concerned, that this approach won't let the authors distinguish between ON and OFF subfields. The considered calcium response transient includes both the stimulus onset and offset back to the grey background. For example, for a white patch at a given location in visual space, the calcium transient would represent an integrated signal of both, the stimulus onset (in this case light increment) and stimulus offset (a light decrement). Given the slow kinetics of GCaMP6s, ON responses can therefore not be precisely distinguished from OFF responses. This has major consequences on the interpretability of the results of clustering of ON and OFF RFs and could very well be a reason for overlapping ON and OFF RF subfields, defined as "complex cells" by the authors. We suggest to instead present each stimulus frame for a longer duration and only take the frames during stimulation into account for RF extraction, in order to ensure precise extraction of the RF subfield polarity/sign.

A brief stimulus pulse (where the width is small compared to the dynamics of the entire dynamics of cellular response) results in an estimate of the **impulse** response of the system. ON and OFF cells are readily told apart given their temporal

responses. The figure on the right, shows the dynamics of responses in one cell. The column on the left shows the responses to bright stimuli, while the one on the right shows the responses to a dark stimulus. In this case, we see spatial structure in the ON kernel starting at 2 frames after stimulus onset and peaking at 4-5 frames after. There is no substantial structure in the OFF kernel. In this case the cell has an ON kernel but no significant OFF kernel. We would classify such cell as an ON cell.

Our figures (and calculations) are based on the ON and OFF maps at the time delay corresponding to the time-to-peak for each individual neuron (as opposed to integrating the entire response over time as the reviewer suggests). This was explained in the methods under the “Calculation of ON/OFF maps” subsection. So, in essence, we are already doing what the reviewer asked by “take the frames during stimulation”.

A stimulus with a longer duration, as proposed by the reviewer (say a 4 sec step), will result in the measurement of a **step** response which could also be used to measure the responses of neurons to increments and decrements in luminance. However, such a method is less efficient than collecting data at 1 stimulus/sec, as one would need to measure for the entire 4 sec and the additional delay to allow the responses to return to baseline.

5.

To reveal ON/OFF domains, the authors employed a canonical correlation analysis (CCA), linking visual and cortical space. They state “It was convenient, for reasons that will become clear soon,...”, but it does not become clear why this analysis was needed. They should elaborate on the use of CCA. For example, what does it mean if clustering is observed in a space that was optimized for maximum correlation?

Please provide intuition on what canonical cortical/visual space is/means. Also, please show CCA weight distributions for each vector.

Why was CCA used in the first place? Were the actual domains in cortical space to diffuse or noisy to become clearly visible? Was there too much variance between the few (6) mice imaged to draw firm conclusions? Ideally, one would like to actually see the domains in the cortex. For this, it might be useful to focus on layer 4, since this is where one would expect the clearest segregation. The paper does not even attempt to analyze the layer specific organization of ON/OFF domains. Instead, it is stated

that “The inclusion of cortical depth (x_3) allowed us to compensate for slight departures of the objective from the surface normal”, but it is not explained how this is done.

A similar question was brought up by reviewer #1. Please see our response to reviewer #1, points #1 and #2. In particular, we hope to have clarified that ON/OFF domains can indeed be demonstrated in the native space.

The reviewer also asks how CCA can compensate for a tilt in the objective. Perhaps the easiest way to explain this is to discuss a simpler situation where we are imaging a two-dimensional piece of cortex (with width and depth) as shown in the diagram on the right. Suppose the cortex is organized so that the azimuth of receptive fields changes linearly along the width of the cortex (\hat{x}_1). The azimuth is assumed constant along cortical columns. Because our sample is slightly tilted with respect to the cortex, our data are represented by the position of cells in the imaging plane and their depth (x_1, x_2). Due to the tilt, the azimuth will change with both x_1 and x_2 , as moving in depth along x_2 means we are also moving a bit along \hat{x}_1 . To correct for this, CCA finds the best linear transformation of (x_1, x_2) such that the result is maximally correlated with azimuth. The result will be one that projects the points to the axis \hat{x}_1 . Note that this is just one number, so we are mapping a dataset in 2 dimensions to 1. It is meaningless to ask for another axis (we have only the azimuth to correlate with). The analogy extends to the case with a dataset with 3 dimension (cortical space) and 2 dimensions (azimuth and elevation, visual space). Here, both datasets will be mapped to a 2-dimensional space.

6. The maps shown in Fig.1 e, f contain large regions with only OFF responses. What is the explanation for this large-scale uneven distribution? Further, there are gaps in the maps, probably due to blood vessels. How do these interfere with the detection of ON/OFF clusters? What are the actual numbers of ON and OFF RFs?

Regions containing an over-abundance of OFF responses represent OFF domains, while regions containing an over-abundance of ON responses represent ON domains. Blood vessels will interfere with the detection of **both** types of cells at those locations. This is the reason our controls consist of relabeling the neurons randomly without changing their positions (as opposed to the classical complete spatial randomness (CSR) baseline in the analysis of spatial point processes). See, for example, the discussion here <https://onlinelibrary.wiley.com/doi/full/10.1111/j.1541-0420.2006.00683.x> The actual numbers of ON and OFF cells in each dataset are now included in the table included in the Methods.

7. The arguments on the distinction between different models of how ON/OFF domains are generated (Fig.2 b-d) are hard to follow. In particular, are the authors confident that given the rather indirect determination of neuronal activity (GCaMP6s, then spike inference), they can exclude a contribution of “synaptic modulation”? The latter phrase is also misunderstandable; I guess “synaptic strengths” captures it better.

Indeed, by “synaptic modulation” we mean “synaptic strength” modulation. That is, that the strength of the drive of ON cells into ON domains is stronger than the drive of ON cells into OFF domains (and vice-versa). We clarify this now in the text. And we agree with the reviewer that such estimates do not represent a direct measure of synaptic strength, but an estimate based on our kernel estimates. We have added caveats in the discussion and propose ways in which the models could be tested in a more direct way.

8. Some of the figure plots miss crucial information; please check for correct and informative axis labels, and for scale bars. It is very unclear what the color code in Fig.1 c is.

See our reply to reviewer #1, minor point #2. We have added additional explanations to all the Figure captions.

9. The authors state that “ON/OFF domains are a critical feature of thalamocortical connectivity”. While this is certainly true for some species, they do not actually show this for the mouse, since they do not record from input fibers, and since it is not clear to which degree their findings hold for layer 4, where thalamic axons mainly terminate,

The reviewer correctly states that we are not directly recording from afferents. In our interpretation of the data, we are assuming that cells that which show single ON or OFF subregions are likely to be dominated by individual afferents. We have now stated this assumption in the text. To continue dissecting the origin of ON/OFF domains, we are now performing dual color imaging recording both the responses of cortical cells and thalamic boutons. We hope to be able to refine our working hypotheses based on those data.

10. The very relevant paper by Smith and Häusser is cited, but just in passing. It should be given credit in the conclusions, too.

Fixed.

Reviewer #3 (Remarks to the Author):

It is thought that On/Off domains of receptive field in primary visual cortex (V1) are derived from thalamocortical afferents, which provide for the columnar cortical architecture of orientation maps observed in carnivores and primates. The paper by Tring, Duan and Ringach shows that On and Off receptive field domains in layer 2/3 of mouse V1 form distinct 100 um-wide clusters. Spatial clustering is an unexpected feature of mouse V1, whose orientation tuned neurons are randomly distributed like salt-and-pepper. The newly found order is intriguing and adds to the emerging evidence of a modular architecture of mouse V1.

1) While the evidence for spatial clustering is quite convincing. It is important to show the distribution of these clusters across V1. Figures 1e and f take a stab at this, but it is difficult to extract specific information from the illustrations.

We hope that our reply to Reviewer #1, point #1, which includes the distribution of ON/OFF domains in native, cortical space, gives the reviewer a better sense of how the data look in cortical coordinates.

Studying the distribution of ON/OFF domains across the entirety of V1 is challenging since we need sufficient magnification to record signals from individual cells, which limits the field of view we can obtain. In principle, we could have run sessions across multiple days by recording different parts of V1 on the same animal and stitched the results together, but this was not done.

We are presently conducting wide-field imaging and 1p excitation of GCaMP in V1 to see if we can recover ON/OFF domains across the entirety of V1, as recently done in the monkey using intrinsic imaging by the Callaway laboratory.

2) Figure 1d and 2a show that the receptive field of a single simple cell representing a given point in visual space is made-up from a pair of On and Off subfields. The authors may consider extending the analysis to the cortical point image, which would give a more complete picture of the number and distribution of On and Off subfields and their possible relationship to previously reported spatially clustered patterns in mouse V1.

Yes, we indeed tried to study the point-spread function by looking at the location of cells in the cortex that respond to luminance increases and decreases at one point in the visual field. However, those analyses turned to be somewhat challenging due to areas of occlusion (generated by blood vessels) present in various parts of the imaging field. Note we will be sharing the kernels and cell segmentations for all our experiments, so members of the community will be in the position to re-analyze the same dataset using alternative techniques.

3) The authors reason that the spatial clustering of On and Off subfields may be related to orientation selectivity. I have difficulties to understand how the apparent order found in their study squares with the disorderly distribution of orientation tuned neurons in mouse V1.

The variability of orientation tuning in the population is large (Fig 4b), causing a disorderly distribution of preferred orientation noted by the reviewer. However, the modeling shows that the receptive fields of all these cells arise from the pooling of a small, local population of ON/OFF signals (Fig 4f,g). In addition, the fluctuations in the relative density of ON/OFF cells, which defines orientation domains, correlates with the average simple cell RF (Fig 4a).

4) In Figure 1d, the panels are too small, making it difficult to extract RF size and spatial displacement of ON and Off subregions. Please provide scale for deg of visual field. How large are the receptive fields of simple and complex cells?

Following the reviewer's suggestion, we now report the sizes of ON and OFF subregions based on the standard deviation of their Gaussian fits. The distribution for these two classes of neuro is shown on the right. ON cells were 13.6 ± 2.7 deg and OFF cells were 12.8 ± 2.4 deg in size.

5) In Figure 1e, how many On and Off neurons/10 deg?

We now report the total number of ON and OFF cells in the populations for each experiment (see new table in Methods). Note that the figure the reviewer is asking will depend heavily on how many planes we sample in each case. However, based on the relative proportion of cells, we can state that the density of OFF cells is about twice as much as ON cells. This is now reported in the text.

6) I recommend using consistent language for domain, subregion, and On/Off maps throughout the text.

Thank you. We checked the manuscript and the terms appear to be used consistently throughout.

7) In Figure 1g, there is a discrepancy of correlation of the second co-variances $p=0.76$ (text) and $p=0.81$ (Figure 1g).

Thank you for catching this. The values reported in the Figure were correct. We fixed the text.

8) In Figures 1f, g, Please provide scales for cortical distance (um) and space (deg). It is difficult to extract the center-to-center distances between On and Off cells, and to get a sense how many On and off subfields are contained within a cortical region representing a specific location in space.

Figures 1fg plot the data in canonical cortical and visual space which lack units, as the distributions in these spaces are normalized to have unit variance. In our reply to reviewer #1 above we show ON/OFF domains in the native visual space, and we now report the average size of ON/OFF domains based on such analyses.

9) In Figures 1e, f, please specify in which part of the visual field the recordings were taken and provide axes for elevation and azimuth.

Please see our reply to reviewer #2 point #2, as s/he brought up a similar concern. We now provide the average location of the RFs in the Methods.

10) In Figure 2a, the title should indicate that these are probability distributions. Maybe I do not read the plots correctly, but it would be helpful to add scales for distance and visual angle.

These panels represent the fluctuations in the distributions plotted in canonical visual space. As stated in the figure legend, all axes range from -2.5 to 2.5. We have modified the title of the figure to clarify we are looking at fluctuations in relative density.

11) Page 4/para 3/line 2. Shouldn't it read On/Off domain (subregions) centers instead of receptive field centers?

We clarified this passage.

12) In Figures 3a, b, f, please provide scales for cortical distance and visual space.

Done

13) In the legend of Figure 3e please replace by "3" by "e".

Thank you! Done.

14) Page 6/para 1, I suggest to improve the last sentence. Clearly indicate that the receptive field is composed of 100 um-wide On and Off clusters.

We now report the average size of ON/OFF domains

15) Page 6/para 4, please revise 2nd to last sentence.

Done

REVIEWER COMMENTS

Reviewer #1 (Remarks to the Author):

The authors have fully addressed the comments from this reviewer.

Reviewer #2 (Remarks to the Author):

ON/OFF domain shape receptive fields in mouse visual cortex

Elaine Tring, Konnie Duan, Dario L. Ringach

The authors have addressed many of our concerns, and the paper has been improved. There are, however, still a number of points which are not really covered in the revised version. In general, while their rebuttal letter has helped clarifying several issues, a number of these have not been incorporated into the revised manuscript, such that readers might still find it difficult to follow the logic.

1. We are still not convinced why CCA is really needed. Figure 1 should put much more emphasis on raw data, clusters should be shown in native space (use Supplementary Figures for more raw data). Please also provide examples of calcium transients in this Figure, single trials and average, for individual ON and OFF stimuli. Also, Figure 1 still misses proper axis ticks & labels (especially for native space). Finally, since the authors did map retinotopy, they should draw an outline of V1 into cortical space.
2. The authors chose to lump together data from different cortical depth levels. While this might increase signal-to-noise, this increase only very indirectly indicates the presence of a columnar organization. The paper would profit a lot from directly showing such columns. I.e. testing whether patches found at different depths are in register, using cross-correlation.
3. The basic finding of ON OFF patches is interesting enough. The paper does not gain from (but rather loses by) adding a toy modeling part, which is not substantiated by real data and, at this point remains speculative and at best only helps forming hypothesis.
4. Please spend more time on crafting the explanations and transitions, such that the general readership, which does not have access to the rebuttal letter, can gain a better understanding of this important finding. See below for one specific example.

Minor

5. As the retinotopic patches were sampled with different trial numbers, please provide data on which patch-position was shown how often (per mouse), to give a better intuition for the sparse noise sampling.
6. Please provide the distribution ON-OFF retinotopic distances for RFs with both subfields, to provide an intuition for the subfield-relationship
7. Reply to our question 5: Was the correction exclusively done to maximize in azimuth or also elevation?
8. Which cortical layers are the imaged cells from?
9. Please elaborate on the assumption that “cells which show single ON or OFF subregions are likely to be dominated by individual afferents”.
10. Line 61/62: “[...] correlating the responses with the locations of bright and dark stimuli in the stimuli, please replace first “stimuli” with “patches”.
11. Please simplify the explanation in the manuscript text on the correction of the surface normal via CCA, as done nicely in the rebuttal letter (maximize azimuth spread).
12. Line 135..: “An important observation that relies on the alignment of visual and cortical spaces produced by canonical correlation analysis is that fluctuations in fx_{ON} - fx_{OFF} are robustly mirrored by fluctuations in the density of receptive field centers on the visual field”, please clarify whether RF subfield centers or overall RF centers are meant.
13. In the discussion on the three models, how do the authors reconcile the clustered input argument with the fact that dLGN arbors are hundreds of microns in diameter (see Antonini et al. 1999)?
14. It is unclear why the aggregate population RF is different from the average population RF. Is this not the same, only with different axis compression/normalization?
15. Analysis presented in Figure 4: it would be important to conduct it separately for L2/3 and L4, as intuitively one might expect different functional connectivity rules there. So please, again, indicate which layers these cells were recorded from.

Reviewer #3 (Remarks to the Author):

This is a successful revision of an important paper. I have no further comments.

We are glad Reviewers #1 and #3 found that the revised version answered their concerns and questions. Reviewer #2 also found the revised version much improved. At the same time, the reviewer offered additional suggestions to improve the presentation of the material which we consider here.

One main request from Reviewer #2 is that we include some of the figures in our initial rebuttal as part of the manuscript in some form. Briefly, in response, we have:

- a) Added a Fig showing the coarse retinotopic mapping (Fig 1a)
- b) Added a Fig showing the demonstration of ON/OFF domains in native cortical space (Fig 2d), along with Sup Fig 1 showing the analysis for all the datasets.
- c) Added a new Fig 2a, exemplifying the dynamics of a response.
- d) Added Fig 2c, which now includes the distribution of normalized distances between ON and OFF kernels.
- e) Added new calculations (Methods, table) showing the correlation between ON/OFF domains calculated from superficial and deep optical planes, which offers further support of the columnar organization of the domains.

We have also modified the text extensively to clarify some additional points brought up by the Reviewer. We hope these changes and the point-by-point replies and clarifications would make the paper now acceptable for publication.

Reviewer #2 (Remarks to the Author):

ON/OFF domain shape receptive fields in mouse visual cortex

Elaine Tring, Konnie Duan, Dario L. Ringach

The authors have addressed many of our concerns, and the paper has been improved. There are, however, still a number of points which are not really covered in the revised version. In general, while their rebuttal letter has helped clarifying several issues, a number of these have not been incorporated into the revised manuscript, such that readers might still find it difficult to follow the logic.

1. We are still not convinced why CCA is really needed. Figure 1 should put much more emphasis on raw data, clusters should be shown in native space (use Supplementary Figures for more raw data). Please also provide examples of calcium transients in this Figure, single trials and average, for individual ON and OFF stimuli. Also, Figure 1 still misses proper axis ticks & labels (especially for native space). Finally, since the authors did map retinotopy, they should draw an outline of V1 into cortical space.

We are now leading the presentation of the data by showing the calculation of ON/OFF domains in native cortical space as suggested by the reviewer. We have also added a Supp Fig 1 showing these calculations for all the datasets we currently have.

We note we perform a coarse retinotopy using 2p data by splitting the field of view in 9 sectors. This allows us to check the sign of the retinotopy. However, we cannot plot an outline of V1 because the field of view is typically entirely within V1 itself.

2. The authors chose to lump together data from different cortical depth levels. While this might increase signal-to-noise, this increase only very indirectly indicates the presence of a columnar organization. The paper would profit a lot from directly showing such columns. I.e. testing whether patches found at different depths are in register, using cross-correlation.

As requested by the reviewer, we are now presenting new calculations (see table in Methods) which include the correlation coefficient between ON/OFF maps calculated using the superficial and deep optical sections. In all cases but one, the maps are significantly correlated, in agreement with the idea that ON/OFF domains span cortical columns.

3. The basic finding of ON OFF patches is interesting enough. The paper does not gain from (but rather loses by) adding a toy modeling part, which is not substantiated by real data and, at this point remains speculative and at best only helps forming hypothesis.

Here we must disagree. The modeling is the basis for testing the correlations in Fig 4 and the fits explaining the variability of receptive fields in Fig 6.

4. Please spend more time on crafting the explanations and transitions, such that the general readership, which does not have access to the rebuttal letter, can gain a better understanding of this important finding. See below for one specific example.

We have tried to improve the text throughout. We are including the tracked version with the changes and a clean version as well (as the changes were substantial).

Minor

5. As the retinotopic patches were sampled with different trial numbers, please provide data on which patch-position was shown how often (per mouse), to give a better intuition for the sparse noise sampling.

The procedure is random. An individual patch has 10% probability of being white. There are a total of 1500 frames in each experiment. Thus, the distribution of the number of white patches at any one location is binomially distributed with $p=0.1$ and $n=1500$. The same is true for black patches. The mean number of white patches is therefore 150, and the standard deviation is $\sqrt{0.1 * 0.9 * 1500} = 11.6$.

6. Please provide the distribution ON-OFF retinotopic distances for RFs with both subfields, to provide an intuition for the subfield-relationship

We provide a histogram with the distribution of normalized distances in Fig 2c.

7. Reply to our question 5: Was the correction exclusively done to maximize in azimuth or also elevation?

CCA finds a transformation that maximizes the correlation between the linear projections. Because retinotopy is mapped in a columnar fashion across V1 (instead of having a component that changes in depth), it will consider both azimuth and elevation on equal terms.

8. Which cortical layers are the imaged cells from?

Layers 2+3 (85%) and 4 (15%).

9. Please elaborate on the assumption that “cells which show single ON or OFF subregions are likely to be dominated by individual afferents”.

The subregions of cortical, simple cells have a size that compares to the width of a single geniculate center. This has been known since the pioneering work of Hubel and Wiesel. Therefore, only a very small number of inputs can possibly be combined by ON/OFF mono-contrast cells without leading to a substantial increase in RF size. We have clearly stated this is an assumption that we are now in the process of verifying by other means.

10. Line 61/62: “[...] correlating the responses with the locations of bright and dark stimuli in the stimuli, please replace first “stimuli” with “patches”.

Thank you. Done.

11. Please simplify the explanation in the manuscript text on the correction of the surface normal via CCA, as done nicely in the rebuttal letter (maximize azimuth spread).

We have expanded on the explanation, but without the figure in the rebuttal.

12. Line 135.: “An important observation that relies on the alignment of visual and cortical spaces produced by canonical correlation analysis is that fluctuations in fx_{ON} - fx_{OFF} are robustly mirrored by fluctuations in the density of receptive field centers on the visual field”, please clarify whether RF subfield centers or overall RF centers are meant.

Thank you. We rephrased this section.

13. In the discussion on the three models, how do the authors reconcile the clustered input

argument with the fact that dLGN arbors are hundreds of microns in diameter (see Antonini et al. 1999)?

That's a good question. Such anatomical data always represents an upper bound on convergence at a single cortical site. Our functional data suggests that many of these thalamocortical synapses are very weak and, instead, only a handful of inputs are relevant. See the discussion in DL Ringach, "Sparse thalamocortical convergence", Curr Bio, 2021.

14. It is unclear why the aggregate population RF is different from the average population RF. Is this not the same, only with different axis compression/normalization?

Note these averages are based on completely different populations. The average μ_s is the mean of all **simple cell** receptive fields. Instead, $\mu_{on} - \mu_{off}$, is the difference between the average of mono-contrast ON cells and mono-contrast OFF cells. In principle, there is no reason why they would be similar. However, the biased-input model predicts such relationship.

15. Analysis presented in Figure 4: it would be important to conduct it separately for L2/3 and L4, as intuitively one might expect different functional connectivity rules there. So please, again, indicate which layers these cells were recorded from.

Most of the neurons in a volume are from L2/3 (~80-85%) and a minority from L4 (~15-20%). We do not currently have sufficient data to perform these analysis only in L4.

REVIEWERS' COMMENTS

Reviewer #2 (Remarks to the Author):

The majority of our concerns were addressed, and we thank the authors for the explanations provided in their rebuttal letter.

Overall, the revised manuscript has been improved a lot, and the arguments have been spelled out much clearer now. Reader will have an easier time to follow the logic. The new analysis demonstrating the presence of a columnar organization for ON/OFF is very exciting and solidifies the basic finding of ON/OFF domains.

We still have a number of minor but nonetheless relevant comments. We think putting these final touches will turn the manuscript into a really nice paper. In several of these points we simply ask the authors to add details given in their rebuttal letter also to the actual manuscript. We also ask for more scale bars.

1. Rebuttal question 5: Please add the standard deviation of patch-presentation per location (11.6) to the methods.
2. Rebuttal questions 8 & 15: Please consolidate the exact fractions of L2/3 & L4 cells, add them to the methods/text and please explain/comment on the exact criterion/methodology used to determine the L2/3-L4 border, which remains elusive.
3. Consolidate the correct writing: um to μm (e.g. in Supplementary Figure 1).
4. Please provide a figure legend for Supplementary Figure 1.
5. Line 61: please change "sparse-noise" to "locally sparse noise" to more accurately capture the visual stimulus.
6. Line 87: the phrase "(projecting out depth)" is unclear, please re-phrase (the authors probably mean 'ignoring depth?').
7. ON/OFF column analysis: Please add a more detailed description on how the correlation analysis was performed, and on which data exactly. Please add a legend explaining the meaning of the asterisks indicating significance in the "layers correlation" column of the methods table. Incidentally, we are very surprised that this crucial bit of information on the columnar organization is hidden in a table, hidden in the methods.
8. Line 114: "such biased-input model" should read "such a biased-input model".
9. Line 256: "ON/OFF domains is an important feature" should read "ON/OFF domains are an important feature".
10. Figure 1a, top: please add scalebar to two-photon overview.
11. Figure 1d, top: please add scalebar to ROI overview.

12. Figure 1d, bottom: please add amplitude-scalebar to spike inferred traces.
13. Figure 2a-b: please add scalebars to RF visual space.
14. Figure 2d & Supplementary Figure 1: Please use a different colorscale, avoiding blue points on a blue background, this way avoiding exaggerating the actual degree of clustering.

We thank the reviewer and the Editor for these final comments. Our point-by-point replies follow.

REVIEWERS' COMMENTS

Reviewer #2 (Remarks to the Author):

The majority of our concerns were addressed, and we thank the authors for the explanations provided in their rebuttal letter.

Overall, the revised manuscript has been improved a lot, and the arguments have been spelled out much clearer now. Reader will have an easier time to follow the logic. The new analysis demonstrating the presence of a columnar organization for ON/OFF is very exciting and solidifies the basic finding of ON/OFF domains.

Thank you.

We still have a number of minor but nonetheless relevant comments. We think putting these final touches will turn the manuscript into a really nice paper. In several of these points we simply ask the authors to add details given in their rebuttal letter also to the actual manuscript. We also ask for more scale bars.

1. Rebuttal question 5: Please add the standard deviation of patch-presentation per location (11.6) to the methods.

Done.

2. Rebuttal questions 8 & 15: Please consolidate the exact fractions of L2/3 & L4 cells, add them to the methods/text and please explain/comment on the exact criterion/methodology used to determine the L2/3-L4 border, which remains elusive.

The transgenic mice used do not express a particular marker in L4. Thus, we can only infer the approximate number of cells in L4 in our population by looking at the depth of each optical section and the average width of cortical layers in the mouse. We considered planes starting at 300um below the surface to be in L4 (see Niell & Stryker, 2008). Using such an approach, we estimate about 25% of our cells are in L4. We have added this information in the Methods.

3. Consolidate the correct writing: um to μm (e.g. in Supplementary Figure 1).

Thank you. Done.

4. Please provide a figure legend for Supplementary Figure 1.

Done.

5. Line 61: please change “sparse-noise” to “locally sparse noise” to more accurately capture the visual stimulus.

The stimulus is sparse in the sense that, on average, the likelihood of a stimulus being present in any one location is 20%. Note that we do not enforce an exclusion zone around each stimulus as done in the “locally sparse” stimulus by the Janelia team (https://observatory.brain-map.org/visualcoding/stimulus/locally_sparse_noise). So, to avoid confusion, we prefer to refer to our stimuli simply as sparse.

6. Line 87: the phrase “(projecting out depth)” is unclear, please re-phrase (the authors probably mean ‘ignoring depth’?).

Done.

7. ON/OFF column analysis: Please add a more detailed description on how the correlation analysis was performed, and on which data exactly. Please add a legend explaining the meaning of the asterisks indicating significance in the “layers correlation” column of the methods table. Incidentally, we are very surprised that this crucial bit of information on the columnar organization is hidden in a table, hidden in the methods.

We have added a caption to the table with the required information.

8. Line 114: “such biased-input model” should read “such a biased-input model”.

Thank you. Fixed.

9. Line 256: “ON/OFF domains is an important feature” should read “ON/OFF domains are an important feature”.

Fixed.

10. Figure 1a, top: please add scalebar to two-photon overview.

Fixed.

11. Figure 1d, top: please add scalebar to ROI overview.

Fixed.

12. Figure 1d, bottom: please add amplitude-scalebar to spike inferred traces.

The shape of the deconvolve curve has arbitrary units. See discussion in Suite2p release 0.7.2, section 8.2. <https://suite2p.readthedocs.io/en/latest/FAQ.html>

13. Figure 2a-b: please add scalebars to RF visual space.

Done

14. Figure2d & Supplementary Figure 1: Please use a different colorscale, avoiding blue points on a blue background, this way avoiding exaggerating the actual degree of clustering.

We feel the colormap is sufficiently different from the points. All the points are clearly visible and easily discriminable from the background. Moreover, the panel on the right, shows the difference in the density of ON and OFF cells without the points and with significance level sets.